

# 1   Daytime formation of nitrous acid at a coastal remote site in
# 2   Cyprus indicating a common ground source of atmospheric
# 3   HONO and NO

Hannah Meusel, Uwe Kuhn[1], Andreas Reiffs[2], Chinmay Mallik[2], Hartwig Harder[2], Monica
Martinez[2], Jan Schuladen[2], Birger Bohn[3], Uwe Parchatka[2], John N. Crowley[2], Horst Fischer[2],
Thorsten Hoffmann[4], Ruud Janssen[2,5], Oscar Hartogensis[6], Michael Pikridas[7], Mihalis
Vrekoussis[7,8,9], Efstratios Bourtsoukidis[2], Bettina Weber[1], Jos Lelieveld[2], Jonathan Williams[2],
Ulrich Pöschl[1], Yafang Cheng[1], Hang Su[1]
[1]Max Planck institute for Chemistry, Multiphase Chemistry Department, Mainz, Germany
[2]Max Planck Institute for Chemistry, Atmospheric Chemistry Department, Mainz, Germany
[3]Institute for Energy and Climate Research (IEK-8), Research Center Jülich, Jülich, Germany
[4]Johannes Gutenberg University, Inorganic and Analytical Chemistry, Mainz, Germany
[5]MeteoGroup, Wageningen, Netherlands
[6]Wageningen University and Research Center, Meteorology and Air Quality, Wageningen, Netherlands
[7]Cyprus Institute, Energy, Environment and Water Research Center, Nicosia, Cyprus
[8] Institute of Environmental Physics and Remote Sensing – IUP, University of Bremen, Germany
[9] Center of Marine Environmental Sciences – MARUM, University of Bremen, Germany
*Correspondence to:* Hang Su (h.su@mpic.de)
**Abstract.** Characterization of daytime sources of nitrous acid (HONO) is crucial to understand atmospheric
oxidation and radical cycling in the planetary boundary layer. HONO and numerous other atmospheric trace
constituents were measured on the Mediterranean island of Cyprus during the CYPHEX campaign (CYPHEX =
CYprus PHotochemical EXperiment) in summer 2014. Average volume mixing ratios of HONO were 35 pptv (± 25
pptv) with a HONO/NO$_x$ ratio of 0.33, which was considerably higher than reported for most other rural and urban
regions. Diel profiles of HONO showed peak values in the late morning (60±28 pptv around 09:00 local time), and
persistently high mixing ratios during daytime (45±18 pptv) indicating that the photolytic loss of HONO is
compensated by a strong daytime source. Budget analyses revealed unidentified sources producing up to 3.4 x 10$^6$
molecules cm$^{-3}$ s$^{-1}$ of HONO and up to 2.0 x 10$^7$ molecules cm$^{-3}$ s$^{-1}$ NO. Under humid conditions (RH >70%), the
source strengths of HONO and NO exhibited a close linear correlation (R²=0.78), suggesting a common source that
may be attributable to emissions from microbial communities on soil surfaces.
**1 Introduction**
Nitrous acid (HONO) is an important component of the nitrogen cycle being widely spread in the environment.
Either in its protonated form (HONO or HNO$_2$) or as nitrite ions (NO$_2^-$) it can be found in the gas phase, on aerosol-
particles, in clouds and dew droplets but also in soil, sea-water and sediments (Foster et al., 1990; Rubio et al., 2002;
Acker et al., 2005 and 2008; Bianchi et al., 1997). It plays a key role in the oxidizing capacity of the atmosphere, as
it is an important precursor of the OH radical, which initiates most atmospheric oxidations. OH radicals react with
pollutants in the atmosphere to form mostly less toxic compounds (e.g. CO + OH → CO$_2$ + H$_2$O; Levy, 1971).





Volatile organic compounds (VOCs) react with OH contributing to formation of secondary aerosols (SOA), which
can serve as cloud condensation nuclei CCN (Arey et al., 1990; Duplissy et al., 2008). Furthermore OH oxidizes $SO_2$
to $H_2SO_4$, which condense subsequently to form aerosol particles (Zhou et al. 2013). In this way HONO has an
indirect effect on the radiative budget and climate. In the first 2-3 hours following sunrise, when OH production from
other sources (photolysis of $O_3$ and formaldehyde) is relatively low, photolysis of HONO can be the major source of
OH radicals as HONO concentrations may be high after accumulation during night time (Lammel and Cape, 1996;
Czader et al., 2012; Mao et al., 2010). On average up to 30% of the daily OH budget in the boundary layer is
provided by HONO photolysis (Alicke et al., 2002; Kleffmann et al., 2005; Ren et al., 2006), but has been reported
as high as 56% (Ren et al., 2003) with ambient HONO mixing ratios ranging from several pptv in rural areas up to a
few ppb in highly polluted regions (Acker et al., 2006a and 2006b; Costabile et al., 2010; Li et al., 2012; Michoud et
al., 2014; Spataro et al., 2013; Su et al. 2008a; Zhou et al., 2002a).
In early studies, atmospheric HONO was assumed to be in a photostationary state during daytime controlled by the
gas phase reaction of NO and OH (R1) and two loss reactions which are the photolysis (R2) and the reaction with
OH (R3).
$$OH + NO \rightarrow HONO \qquad (R1)$$
$$HONO \xrightarrow{hv\ (300-405\ nm)} OH + NO \qquad (R2)$$
$$HONO + OH \rightarrow NO_2 + H_2O \qquad (R3)$$
However, field measurements in remote and rural locations, as well as urban and polluted regions found several
times higher daytime HONO concentrations than model predictions, suggesting a large unknown source (Kleffmann
et al., 2003 and 2005; Su et al., 2008a; Soergel et al., 2011a; Su et al., 2011; Michoud et al., 2014; Czader et al.,
2012; Wong et al., 2013; Tang et al., 2015; Oswald et al., 2015) even after considering direct emission of HONO
from combustion sources (Kessler and Platt, 1984; Kurtenbach et al., 2001). Heterogeneous reactions on aerosols
have been proposed as an explanation for the missing source. The hydrolysis (R4, Finlayson-Pitts et al., 2003) and
redox reactions of $NO_2$ have been intensively investigated on different kinds of surface such as fresh soot, aged or
organic-coated particles (Amman et al., 1998; Arens et al., 2001; Aubin et al., 2007; Bröske et al., 2003; Han et al.,
2013; Kalberer et al., 1999; Kleffmann et al., 1999; Kleffmann and Wiesen, 2005; Lelievre et al., 2004). Minerals
like $SiO_2$, $CaCO_3$, $CaO$, $Al_2O_3$, and $Fe_2O_3$ showed a catalytic effect on the hydrolysis of $NO_2$ (Kinugawa et al., 2011;
Liu et al., 2015; Wang et al., 2003; Yabushita et al., 2009). Different kind of surfaces (humic acid and other organic
compounds, titanium dioxide, soot) can be photochemically activated which leads to enhanced $NO_2$ uptake and
HONO production (R5,George et al., 2005; Langridge et al., 2009; Monge et al., 2010; Ndour et al., 2008; Ramazan
et al., 2004; Stemmler et al., 2007; Kebede et al., 2013). The photolysis of particulate nitric acid ($HNO_3$), nitrate
($NO_3^-$) and nitro-phenols ($R-NO_2$) lead to HONO formation as well (Baergen and Donaldson, 2013; Bejan et al.,
2006; Ramazan et al., 2004; Scharko et al., 2014; Zhou et al., 2003; Zhou et al., 2011). But these reactions cannot
account for the HONO levels observed during daytime (Elshorbany et al., 2012).
$$2\,NO_2 + H_2O \rightarrow HONO + HNO_3 \qquad (R4)$$





$$\text{surface} \xrightarrow{h\nu} e^- \xrightarrow{NO_2} NO_2^- \xrightarrow{H_2O} HONO + OH^- \tag{R5}$$

On the other hand, soil nitrite, either biogenic or non-biogenic, has been suggested as an effective source of HONO (Su et al., 2011; Oswald et al., 2013). Depending on soil properties such as pH and water content and according to Henry´s law HONO can be released (Donaldson et al., 2014b; Su et al., 2011). This is consistent with field flux measurements showing HONO emission from the ground rather than deposition as is the case for $HNO_3$ (Harrison and Kitto, 1994; Kleffmann et al., 2003; Ren et al., 2011; Stutz et al., 2002; VandenBoer et al., 2013; Villena et al., 2011; Wong et al., 2012 and 2013; Zhou et al., 2011). In a recent study, Weber et al. (2015) measured large HONO- and NO-emissions from dryland soils with microbial surface communities (so-called biological soil crusts).

Several field studies also show a correlation of the unknown HONO source with solar radiation or the photolysis frequency of $NO_2$ $J_{NO2}$ (Su et al., 2008a; Sörgel et al., 2011a; Wong et al., 2012; Costabile et al., 2010; Michoud et al., 2014, Oswald et al., 2015). This correlation can be explained either by the aforementioned photosensitized reactions or by temperature-dependent soil-atmosphere exchange (Su et al., 2011). According to Su et al. (2011), the release of HONO from soil surfaces is controlled by both the soil (biogenic and chemical) production of nitrite and the gas-liquid phase equilibrium. The solubility is strongly temperature-dependent, resulting in a higher HONO emission during noon time and high radiation $J_{NO2}$ periods, and lower HONO emissions or even HONO deposition during the nighttime as further confirmed by VandenBoer et al. (2015). This temperature dependence not only exists for equilibrium over soil solution but also exists for adsorption/desorption equilibrium over dry and humid soil surfaces (Li et al., 2016).

In this study we measured HONO and a suite of other atmospherically relevant trace gases in a coastal area on the Mediterranean Island Cyprus in summer 2014. Due to low local anthropogenic impact and low NOx levels in aged air masses, but high solar radiation, this is an ideal site to investigate possible HONO sources and to gain a better understanding of HONO chemistry.

## 2 Instrumentation

HONO was measured with a commercial Long Path Absorption Photometry instrument (effective light path 1.5 m, LOPAP, Quma, Wuppertal, Germany). LOPAP has a collecting efficiency of >99% for HONO and a detection limit of 4 pptv at a time resolution of 30s. To avoid potential interferences induced by long inlet lines and heterogeneous formation or loss of HONO on the inlet walls, respectively (Kleffmann et al., 1998; Zhou et al., 2002b; Su et al., 2008b), HONO was collected by a sampling unit installed directly in the outdoor atmosphere, i.e., placed on a mast at a height of 5.8 meters above ground installed at the edge of a laboratory container. Furthermore, the LOPAP has two stripping coils placed in series to reduce known interfering signals (Heland et al., 2001). In the first stripping coil HONO is quantitatively collected. Due to the acidic stripping solution interfering species are collected less efficiently but in both channels. The true concentration of HONO is obtained by subtracting the inferences quantified in the second channel (in this study average 1 pptv, at most 5 pptv) from the total signal obtained from the first channel. For a more detailed description of LOPAP, see Heland et al. (2001). This correction of chemical interferences ascertained excellent agreement with the (absolute) DOAS measurements, both in a smog chamber and under urban atmospheric conditions (Kleffmann et al., 2006). A possible interference from peroxynitric acid ($HNO_4$)



has been proposed (Liao et al., 2006; Kerbrat et al., 2012; Legrand et al., 2014), but this will be insignificant at the
high temperatures during CYPHEX, at which $HNO_4$ is unstable. The stripping coils are temperature controlled by a
water-based thermostat and the whole external sampling unit is shielded from sunlight by a small plastic housing.
The reagents were all high purity grade chemicals, i.e., hydrochloric acid (37%, for analysis; Merck), sulfanilamide
(for analysis, >99%; AppliChem) and N-(1-naphthyl)-ethylenediamine dihydrochloride (for analysis, >98%;
AppliChem). For calibration Titrisol® 1000 mg $NO_2^-$ ($NaNO_2$ in $H_2O$; Merck) was diluted to 0.0015 and 0.005 mg/L
$NO_2^-$. For preparation all solutions and for cleaning of the absorption tubes 18 MΩ $H_2O$ was used.
NO and $NO_2$ measurements were made with a modified commercial chemiluminescence Detector (CLD 790 SR)
originally manufactured by ECO Physics (Duernten, Switzerland). The two-channel CLD based on the
chemiluminescence of the reaction between NO and $O_3$ was used for measurements of NO and $NO_2$. $NO_2$ was
measured as NO using a photolytic converter from Droplet Measurement Technologies, Boulder USA. In current
study, data were obtained at a time resolution of 5 seconds. The CLD detection limits (determined by continuously
measuring zero air at measuring site) for NO and $NO_2$ measurements were 5 pptv and 20 pptv, respectively for an
integration period of 5 s. $O_3$ was measured with a standard UV photometric detector (Model 49, Thermo
Environmental Instruments Inc.) with a detection limit of 1 ppb. Data are reported for an integration period of 60 s.
The total uncertainties (2σ) for the measurements of NO, $NO_2$ and $O_3$ were determined to be 20%, 30% and 5%,
respectively, based on the reproducibility of in-field background measurements, calibrations, the uncertainties of the
standards and the conversion efficiency of the photolytic converter (Li et al., 2015).
OH and $HO_2$ radicals were measured using the HydrOxyl Radical measurement Unit based on fluorescence
Spectroscopy (HORUS) setup developed at the Max Planck Institute for Chemistry (Mainz, Germany). HORUS is
based on laser induced fluorescence- fluorescence assay by gas expansion (LIF-FAGE) technique, wherein OH
radicals are selectively excited at low pressure by pulsed UV light at around 308 nm, and the resulting fluorescence
of OH is detected using gated microchannel plate (MCP) detectors (Martinez et al., 2010; Hens et al., 2014). $HO_2$ is
estimated by converting atmospheric $HO_2$ into OH using NO, and detecting the additional OH formed. The
instrument is calibrated by measuring signals from known amounts of OH and $HO_2$ generated by photolysis of water
vapor in humidified zero air.
Photolysis frequencies were determined using a spectroradiometer (Metcon GmbH) with a single monochromator
and 512 pixel CCD-array as detector (275-640 nm). The thermostatted monochromator/detector unit was attached via
a 10 m optical fiber to a 2-Π integrating hemispheric quart dome. The spectroradiometer was calibrated prior to the
campaign using a 1000 W NIST traceable irradiance standard. J-values were calculated using molecular parameters
recommended by the IUPAC and NASA evaluation panels (Sander et al., 2011; IUPAC, 2015). The J-value for
HONO was not corrected for upwelling UV radiation and is estimated to have an uncertainty of ~10 % (Bohn et al.,

33  2008).

Aerosol measurements were also performed during the campaign. In this study particulate nitrate and aerosol surface
data were used. These were detected by high resolution – time of flight – aerosol mass spectrometer (HR-ToF-AMS,
Aerodyne Research Inc., Billerica, MA USA) and scanning mobility particle sizer (SMPS 3936, TSI, Shoreview,
MN USA) and aerodynamic particle sizer (APS 3321, TSI), respectively. The mobility and aerodynamic based size
distributions were combined based on the algorithm proposed by Khlystov et al. (2004).



The volatile organic compounds (VOC) including α-pinene, β-pinene, isoprene, Δ3-carene, limonene and DMS
(dimethyl sulfide) were detected by a commercial Gas Chromatography-Mass Spectrometry (GC-MS) system (MSD
5973; Agilent Technologies GmbH) coupled with an air sampler and a thermal desorber unit (Markes International
GmbH). The VOCs were trapped at 30$^o$C on a low-dead-volume quartz cold trap (U-T15ATA; Markes International
GmbH) filled with two bed sorbent (Tenax TA and Carbograph I). The cold trap was heated to 320$^o$C and the sample
was transferred to a 30m GC column (DB-624, 0.25mm I.D., 1.4μm film; J&W Scientific). The temperature of the
GC oven was programmed to be stable at 40$^o$C for 5mins and then rising with a rate of 5$^o$C/min up to 140$^o$C.
Following, the rate was increased to 40$^o$C/min up to 230$^o$C where it was stabilized for 3min. Each sample was taken
every 45mins and calibrations, using a commercial gas standard mixture (National Physical Laboratory, UK), were
performed every 8-12 samples.
Formaldehyde (HCHO) was measured with a commercial analyzer based on the Hantzsch reaction. The product of
the reaction of HCHO with acetyl-acetone and ammonia absorbs light at 410 nm and fluoresces at 510 nm which is
detected (AL4011, Aerolaser GmbH, Garmisch-Partenkirchen, Germany).
Carbon monoxide was measured by infrared absorption spectroscopy using a room temperature quantum cascade
laser at a time resolution of 1 s. Data are reported as 60 s averages with a total uncertainty of ~10% mainly
determined by the uncertainty of the used NIST standard (Li et al., 2015).
Meteorological parameters (temperature, relative humidity, wind speed and wind direction, pressure, solar radiation,
precipitation) were detected by the weather station Vantage Pro2 from DAVIS.
Besides GC-MS all other operating instruments had time resolutions between 20 s and 5 min. For most analyses in
this study the data were averaged to 10 min. When GC-MS data were included in the evaluation 1 hour averaged data
were used.
**3 Site description**
Cyprus is a 9251 km² island in the South-East Mediterranean Sea (fig. 1). The measuring site was located on a
military compound in Ineia, Cyprus (N 34.9638, E 32.3778), about 600 m above sea level and approximately 5.5 - 8
km from the coast line (in the main wind direction W-SW). The field site is characterized by light vegetation cover,
mainly comprising small shrubs like *Pistacia lentiscus, Sacopoterium spinosum,* and *Nerium oleander,* herbs like
*Inula viscosa* and *Foeniculum vulgare* and few typical Mediterranean trees like *Olea europaea, Pinus* sp.*, and*
*Ceratonia siliqua..* The area within a radius of about 15 km around the station is only weakly populated. Paphos
(88,266 citizens) is located 20 km south of the field site, Limassol (235,000), Nicosia (325,756) and Larnaca
(143,367) are 70, 90 and 110 km in the E-SE, respectively (population data according to statistical service of the
republic of Cyprus, www.cystat.gov.cy, census of population Oct 2011). During the campaign (07.07. - 04.08.2014),
clear sky conditions prevailed and occasionally clouds skimmed the site. No rain was observed, but the elevated field
site was impacted by fog during nighttime and early morning due to adiabatic cooling of ascending marine humid air
masses. Temperature ranged from 18 to 28°C. Within the main local wind direction of SW (fig. 2A) there was no
direct anthropogenic influence resulting in clean humid air from the sea. Analysis of 48-hours back trajectories
showed mainly two source regions of air mass origin (fig. 2B). Approximately half (46%) of the campaign the air





masses came from the West of Cyprus spending most of their time over the Mediterranean Sea prior to arriving at the
site. During the remaining half of the campaign air masses originated from the North of Cyprus, from East European
countries (Turkey, Bulgaria, Rumania, Ukraine and Russia). Westerly air masses have been shown to exhibit lower
concentration of gaseous and aerosol pollutants than the predominant northerly air masses that typically reach the
site (Kleanthous et al., 2014). They spent more time over continental terrestrial surface and were likely to be
additionally affected by biomass burning events detected in East Europe within the measurement periods (FIRMS,
MODIS, web fire mapper, fig. S1). Previous back trajectory studies in the eastern Mediterranean support this
assumption (Kleanthous et al., 2014; Pikridas et al., 2010).
Most of the time the advected air mass was loaded with high humidity as a result of sea breeze circulation. Two
periods of about 4 days with lower relative humidity occurred. These two situations will be contrasted below.
**4 Results**
The concentrations of HONO and other atmospheric trace gases as well as meteorological conditions observed on
Cyprus from $7^{th}$ July 2014 to $3^{rd}$ August 2014 are shown in fig. 3. In general, low trace gas mixing ratios were
indicative of clean marine atmospheric boundary conditions, as pollutants are oxidized by OH during the relatively
long air transport time over the Mediterranean sea (more than 30 h), and without significant impact of direct
anthropogenic emissions.
Ambient HONO mixing ratios ranged from below detection limit (< 4 pptv) to above 300 pptv. Daily average HONO
was 35 pptv (± 25 pptv). The daily average $NO_2$ and NO mixing ratios were 140 ± 115 and 20 ± 35 pptv
respectively, but showed intermittent peaks up to 50 ppbv when sampling air was streamed from the diesel generator
used to power the station, from the access route or the parking lot by local winds (easterly, fig S2). These incidents,
which account for 4% of the campaign time, were classified as local air pollution events and were omitted from
analysis. Mean $O_3$ and CO mixing ratios were 72 ± 12 ppb and 98 ± 11 ppbv respectively. OH radicals ranged from
below detection limit ($1 \times 10^5$ molecules $cm^{-3}$) during nighttime to $8 \times 10^6$ molecules $cm^{-3}$ during daytime. Daytime
$HO_2$/OH ratio ranged from 100 to 150. The mixing ratios of $NO_2$, $O_3$ and CO varied in unison, and were significantly
($p<0.05$) higher during periods when air masses originated from East Europe (brownish bar in fig. 3a lower panel),
indicative of air pollution and shorter transport times compared to western Europe ($NO_2$: Northerly: 144 ± 130 pptv,
westerly: 127 ± 106 pptv; $O_3$: Northerly: 74 ± 11 ppbv, westerly: 66 ± 12 ppbv; CO: Northerly: 101 ± 9 ppbv,
westerly: 90 ± 10 ppbv). In contrast, NO and HONO mixing ratios were slightly higher when air masses came from
Western Europe and over the sea (NO: Northerly: 17 ± 35 pptv, westerly: 20 ± 44 pptv; HONO: Northerly: 32 ± 26
pptv, westerly: 38 ± 22 pptv).
Besides two different air mass origins, two periods with different behaviour of relative humidity were identified
illustrated by blue and yellow boxes in fig. 3(a and b). In both periods we found northerly and westerly air mass
origins. The diel profiles of trace gas mixing ratios and meteorological variables of the humid period (blue box) are
shown in Fig. 4a, the ones of the dry period (yellow box) in Fig 4b. During the drier period HONO concentrations
are stable and low (6 pptv) during night, while mean nighttime HONO mixing ratios during the humid period (fig.
4a) showed an expected slow increase of about 20 pptv (from 20 to 40 pptv), as anticipated from heterogeneous





production and accumulation within a nocturnal boundary layer characterized by a stable stratification and low wind
speed (Acker et al., 2005; Su et al., 2008b; Li et al., 2012). During both periods, but more pronounced in the drier
period, HONO rapidly increased by a factor of 2 within two hours after sunrise and then slowly decreased until
sunset. Similar profiles were also observed for other trace gases like isoprene or DMS which are transported in
upslope winds. Strong HONO morning peaks and high daytime mixing ratios suggest a strong daytime source,
compensating the short atmospheric lifetime (15 min) caused by fast photolysis.
Mean NO mixing ratios were close to the detection limit (2 pptv) during night and increased after sunrise (06:00
local time LT) to mean values of 60 pptv (peak 150 pptv) at 09:00 LT, prior to declining for the rest of the day until
sunset (20:00 LT). In the absence of local NO sources low nighttime values are a result of the conversion of NO to
$NO_2$ by $O_3$ (Hosaynali Beygi et al., 2011). The diel profiles of NO mixing ratios followed closely those of HONO
mixing ratios. This similarity and their dependency on relative humidity are suggestive of a common source for both
reactive nitrogen species.
$NO_2$ mixing ratios were somewhat lower during nighttime, but in general the diel variability remained in a narrow
range between 100 and 200 pptv. Likewise, the diel courses of $O_3$ and CO mixing ratios revealed relatively low
day/night variability in a range of 65-75 and 90-100 ppb, respectively.
**5 Discussion**
Low $NO_x$ conditions at this remote field site in photochemically aged marine air were found to be an ideal
prerequisite to trace yet un-defined local HONO sources. On Cyprus, diel profiles of HONO showed peak values in
the late morning and persistently high mixing ratios during daytime, as has been reported for some other remote
regions (Acker et al., 2006a; Zhou et al., 2007; Huang et al., 2002). This is not the case for rural and urban sites,
where atmospheric HONO mixing ratios are normally observed to continuously build up during nighttime
presumably due to heterogeneous reactions involving $NO_x$ and decline in the morning due to strong
photodissociation (e.g., Elshorbany et al., 2012 and references therein).
The diel HONO/$NO_x$ ratio (fig. 4a+b, third panel) shows consistently high values during the humid period (fig. 4a)
and significant diel variation for the dry case (fig. 4b) with higher values during day. The ratio (average 0.33 and
peak values greater than 2) is higher than that reported for most other regions, suggesting a strong impact of local
HONO sources. Elshorbany et al. (2012) investigated data from 15 different urban and rural field measurement
campaigns around the globe, and came up with a robust representative mean atmospheric HONO/NOx ratio as low
as 0.02. However, high values were observed at remote mountain sites, with mean values of 0.23 (up to ≈0.5 in the
late morning; Zhou et al., 2007) or 0.2–0.4 at remote arctic/polar sites (Li, 1994; Zhou et al., 2001; Beine et al.,
2001; Jacobi et al., 2004; Amoroso et al., 2010). Legrand et al. (2014) observed HONO/$NO_x$ ratios between 0.27 and
0.93 during experiments with irradiated Antarctic snow depending on radiation wavelength, temperature and nitrate
content. Elevated HONO/$NO_x$ ratios at low $NO_x$ levels show the importance of HONO formation mechanisms other
than heterogeneous $NO_x$ reactions.



### 5.1 Nighttime HONO accumulation

Between 18:30 – 7:30 LT HONO has an atmospheric lifetime of more than 45 min and [OH] is low, just about $1 \times 10^5$ molecules $cm^{-3}$, so that the calculation of HONO at photostationary state $[HONO]_{pss}$ (R1-R3) at night is not appropriate. Instead, nighttime HONO concentrations can be estimated due to heterogeneous reaction of $NO_2$ described in Eq. (1). Three studies in different environments from a rural forest region in East Germany (Sörgel et al., 2011b) and a non-urban site in the Pearl River delta, China (Su et al., 2008b) to a urban, polluted site in Beijing (Spataro et al., 2013) found a conversion rate of 1.6% $h^{-1}$.

$$[HONO]_{het} = [HONO]_{evening} + 0.016\ h^{-1}[NO_2]\ \Delta t, \qquad\qquad (Eq.\ 1)$$

$[HONO]_{het}$ denotes the accumulation of HONO by heterogeneous conversion of $NO_2$, $[HONO]_{evening}$ the measured HONO mixing ratio at 18:30 LT, $[NO_2]$ the measured average $NO_2$ mixing ratio between 18:30 and 7:30 LT, $\Delta t$ time span in hours.

Measured and calculated HONO mixing ratios are compared in figure 4 (upper panel). During the humid period, during night the estimated (according Eq. (1), fig. 4a upper panel, grey line) and observed HONO mixing ratios are in good agreement ($R^2 = 0.9$). During the drier period the observed HONO mixing ratios were lower than the ones calculated with a $NO_2$ conversion rate of 1.6% $h^{-1}$. But Kleffmann et al., 2003 found a smaller conversion rate of $6 \times 10^{-7}\ s^{-1}$ (0.22% $h^{-1}$) for rural forested land in Germany which matches better to the observed nighttime HONO concentration during drier period (fig. 4b upper panel, dark grey line).

As already mentioned above, it is apparent that under low RH conditions during night, HONO mixing ratios were much lower than under humid conditions, and HONO morning peaks were most pronounced (compare Fig. 4a and 4b: humid/dry). Both HONO (Donaldson et al., 2014a) and $NO_2$ (Wang et al., 2012; Liu et al., 2015) uptake coefficients have recently been reported to be much stronger for dry soil, or at low RH, respectively, which is in line with HONO on Cyprus being close to the detection limit in nights with low relative humidity. On the other hand, it has been shown on glass and on soil proxies that the yield of HONO formation from $NO_2$ on surfaces is low under dry conditions, but sharply increases at RH >30% (Liu et al., 2015) or >60% (Finlayson-Pitts et al., 2003). On Cyprus the strong morning HONO peaks after dry nights were accompanied by an increase in relative humidity from 40 to 80%. Deposited and accumulated $NO_2$ on dry soil surfaces could be released as HONO at high rates under elevated RH conditions. In contrast, in a humid regime HONO mixing ratios were continuously high during nighttime and showed less pronounced morning peaks, suggesting lower nighttime deposition of $NO_2$ and lower HONO emissions in the morning, respectively.

As morning HONO peak mixing ratios were most pronounced after dry nights on Cyprus, our observations are to some extent contradictory to earlier results that have proposed that dew formation on the ground surface may be responsible for HONO nighttime accumulation in the aqueous phase, followed by release from this reservoir after dew evaporation the next morning (Zhou et al., 2002a, Rubio et al., 2002, He et al., 2006). We cannot rule out that the latter could have contributed to nighttime accumulation of HONO during humid conditions, as we had no means to measure dew formation at the site, and high daytime HONO mixing ratios were observed under all humidity regimes. However, kinetic models of competitive adsorption of trace gases and water onto particle surfaces predict exchange behavior explicitly distinct from the liquid phase (Donaldson et al., 2014a). The nitrogen composition in



thin water films (few water molecular monolayers) is complex, including HONO, NO, $HNO_3$, water–nitric acid
complexes, $NO_2^+$ and $N_2O_4$ (Finlayson-Pitts et al., 2003). With only small amounts of surface-bound water, nitric
acid is largely undissociated $HNO_3$ and is assumed to be stabilized upon formation of the $HNO_3$–$H_2O$ complexes
(hydrates), which have unique reactivity compared to nitric acid water aqueous solutions, where it is dissociated $H^+$
and $NO_3^-$ ions (Finlayson-Pitts et al., 2003). Likewise, HONO formation rates in surface bound water are about four
orders of magnitude larger than expected for the aqueous phase reaction (Pitts et al., 1984).
Diel HONO profiles very similar to those on Cyprus with a late morning maximum and late afternoon/early evening
minimum have been observed at the Meteorological Observatory Hohenpeissenberg, a mountain-top site in Germany
(Acker et al., 2006a) and by Zhou et al. (2007) at the summit of Whiteface Mountain in New York State. For the
latter study, formation of dew could be ruled out as relative humidity was mostly well below saturation. Zhou et al.
(2007) argued that the high HONO mixing ratios during morning and late morning can be explained by mountain up-
slope flow of polluted air from the cities at the foot of the mountain that results from ground surface heating.  On
Cyprus the sea breeze, driven by the growing difference between sea and soil surface temperature, brings air to the
site which interacted with the soil surface and vegetation and is loaded by respective trace gas emissions. This is
endorsed by the simultaneous increase of DMS and isoprene, markers for transportation of marine air and emission
by vegetation. In the late afternoon, when the surface cools, down-welling air from aloft would dominate, being less
influenced by ground surface processes.  Zhou et al. (2007) could show that noontime HONO mixing ratios and
average $NO_y$ during the previous 24-hour period were strongly correlated, much better than instantaneous
$HONO/NO_y$ or $HONO/NO_x$, which is in line with N-accumulation on soil surfaces as discussed above.
**5.2 Daytime HONO budget**
During daytime (7:30 to 18:00 LT, with HONO lifetime being less than 30 min), $[HONO]_{PSS}$, the photostationary
HONO mixing ratios resulting from gas phase chemistry can be calculated according to Eq. (2) (Kleffmann et al.,

23  2005):

$$[HONO]_{PSS} = \frac{k_1[OH][NO]}{k_2[OH] + J_{HONO}}$$  (Eq.2)
where $k_1$ and $k_2$ are the rate constants for the gas phase HONO formation from NO and OH and the loss of HONO by
reaction of HONO and OH, respectively (Atkinson et al., 2004). $J_{HONO}$ is the photolysis frequency of HONO, which
was measured with a spectroradiometer. [NO] is the observed NO mixing ratio. Since OH data were available only
on a few days, diel variations of [OH] were averaged (see fig. S3).
As has been previously established by many other studies (Su et al., 2008b; Michoud et al., 2014; Soergel et al.,
2011a), homogeneous gas-phase chemistry alone fails to reflect observed HONO mixing ratios. Observed daytime
values were up to 30 times higher than calculated based on PSS, indicating strong additional local daytime sources of
HONO. The strength of these sources ($S_{HONO}$) can be calculated by following equation:
$$S_{HONO} = ([HONO]_{measured} - [HONO]_{PSS}) \cdot (k_2[OH] + J_{HONO})$$  (Eq. 3)
In the late morning (around 10:00 LT) the unknown source was at its maximum with peak production rates of up to
$3.4 \times 10^6$ molecules $cm^{-3}$ $s^{-1}$, and a daytime average of about $1.3 \times 10^6$ $cm^{-3}$ $s^{-1}$, which is in good agreement with other



studies at rural sites like a mountain site at Hohenpeissenberg (($3\pm1$) x $10^6$ $cm^{-3}$ $s^{-1}$, at $NO_x \approx 2$ ppbv, Acker et al.,
2006a), a deciduous forest site in Jülich ($3.45\times10^6$ molecules $cm^{-3}$ $s^{-1}$, at $NO \approx 250$ pptv, Kleffmann et al., 2005) and
a pine forest site in South-West Spain $0.74\times10^6$ molecules $cm^{-3}$ $s^{-1}$, at $NO_x \approx 1.5$ ppbv, Sörgel et al., 2011a) but
smaller than at urban sites in Houston ($4\text{-}6\times10^6$ $cm^{-3}$ $s^{-1}$, at $NO_x \approx 6$ppbv, Wong et al., 2012), Beijing ($7\times10^6$ $cm^{-3}$ $s^{-1}$,
at $NO_x \approx 15$ ppbv, Yang et al., 2014) and South China ($5.25 \pm 3.75\times10^6$, at $NO_x \approx 20$ ppbv, Li et al., 2012; or $1\text{-}4\times10^7$
$cm^{-3}$ $s^{-1}$, at $NO_x \approx 35$ ppbv, Su et al., 2008a).
The contributions of gas phase reactions and the heterogeneous reaction of $NO_2$ (conversion rate 1.6% $h^{-1}$) to the
HONO budget are illustrated in fig. 5, exemplary. For both periods the contributions are quiet similar just the
absolute values are different. To compensate the strong loss via photolysis a comparable strong unknown source is
necessary as the heterogeneous $NO_2$ conversion or the gasphase reaction of OH and NO are insignificant.
In polluted regions with moderate to high $NO_x$ concentrations, HONO sources have often been linked with [$NO_2$] or
[$NO_x$] (Acker et al., 2005, Li et al., 2012, Levy et al., 2014, Sörgel et al., 2011a, Wentzel et al., 2010). Under the
prevailing low $NO_x$ conditions during CYPHEX (<250 pptv), correlation analysis (see table 1) of $S_{HONO}$ with $NO_2$
($R^2 = 0.44$) and $NO_2*RH$ ($R^2 = 0.46$) indicate no significant impact of instantaneous heterogeneous formation of
HONO from $NO_2$. Better correlations of $S_{HONO}$ with $J_{NO2}$ ($R^2 = 0.74$) and $J_{NO2}*[NO_2]$ ($R^2 = 0.84$) indicate a photo-
induced conversion of $NO_2$ to HONO as already suggested by George et al. (2005) or Stemmler et al. (2006, 2007).
Other light dependent reactions such as the photolysis of nitrate might additionally contribute to high daytime
HONO. It is unlikely that aerosol surfaces played an important role in heterogeneous conversion of $NO_2$ as the mean
observed aerosol surface concentration was only about 300 $\mu m^2$ $cm^{-3}$. Based on a formula for photo enhanced
conversion of $NO_2$ on humic acid aerosols which was derived by Stemmler et al. (2007) a HONO formation rate of
only $5.1\times10^2$ molecules $cm^{-3}$ $s^{-1}$ can be estimated. Likewise, Sörgel et al. (2015) showed that HONO fluxes from
light-activated reactions of $NO_2$ on humic acid surfaces at low $NO_2$ levels (< 1 ppb and thus comparable to
concentrations observed in this study) saturated at around 0.0125 nmol $m^{-2}$ $s^{-1}$. Therefore heterogeneous aerosol
surface reactions can be neglected as HONO sources at the prevailing low $NO_x$ levels.
Likewise, the nitrate concentrations of highly acidic marine aerosols particulate matter as measured by HR-ToF-
AMS (PM1 fraction, mean 0.075 $\mu g$ $m^{-3}$) were too low to account for significant photolytic HONO production
($1.7\times10^2$ molecules $cm^{-3}$ $s^{-1}$ or 0.01% of $S_{HONO}$) calculated by Eq. (4):

$$S_{photo\_NO_3^-} = [\overline{NO_3^-}] \cdot J_{NO_3^-} \tag{Eq. 4}$$

with $S_{photo\_NO3^-}$ the source strength of HONO by photolysis of nitrate, [$\overline{NO_3^-}$] the mean particulate nitrate
concentration and $J_{NO_3^-}$ the photolysis frequency of nitrate (aq) at noon ($3 \times 10^{-7}$ $s^{-1}$, Jankowski et al., 1999).
Recently an enhancement of the photolysis frequency of particulate nitrate relative to gaseous or aqueous nitrate was
found (Ye et al., 2016). But even with this enhanced rate of $2\times10^{-4}$ $s^{-1}$ not more than $1.1\times10^5$ molecules $cm^{-3}$ $s^{-1}$ (8%
of $S_{HONO}$) HONO would be produced.




**5.3 Common daytime source of HONO and NO**
During CYPHEX, good correlation was found between [HONO] or $S_{HONO}$ and [NO] ($R^2$ = 0.86 and 0.64,
respectively), indicating that both may have a common source. A missing source of NO based on the photostationary
state can be calculated as shown in Eq. (5) and (6).

$$[NO]_{PSS} = \frac{J_{NO_2}[NO_2]+J_{HONO}[HONO]}{k_1[OH]+ k_3[HO_2]+k_4[O_3]+k_5[RO_2]}$$
(Eq. 5)

$$S_{NO} = ([NO]_{measured} - [NO]_{PSS}) \cdot (k_1[OH] + k_3[HO_2] + k_4[O_3] + k_5[RO_2])$$
(Eq. 6)

$k_3$ and $k_4$ are the rate constants for the reaction of NO with $HO_2$ and $O_3$, respectively (Atkinson et al., 2004), $k_5$ is the
rate constant for the reaction of NO and organic peroxy radicals which was assumed to be the same as for the
reaction NO + $CH_3O_2$ (7.7 x $10^{-12}$ $s^{-1}$ at 298K, Ren et al., 2010; Sander et al., 2011). Like [OH] also [$HO_2$] was
measured only on a few days and therefore mean diel data were used (fig. S3). Total [$RO_2$] was estimated to be
maximum 1.6*[$HO_2$] (Ren et al., 2010; Hens et al., 2014). Using a $RO_2/HO_2$ ratio of 1.2 the absolute values of $S_{NO}$
are reduced by 0.3 to 5.5%. The budget analysis for NO for both humidity regimes is illustrated in fig. S4.
For $NO_x$, an unexpected deviation from the PSS, or Leighton ratio, respectively, of clean marine boundary layer air
has been observed previously, invoking a hitherto unknown NO sink, or pathway for NO to $NO_2$ oxidation, other
than reactions with OH, $HO_2$, $O_3$ and organic peroxides (Hosaynali Beygi et al., 2011). On Cyprus, two different
atmospheric humidity regimes can be differentiated. Under dry conditions (RH < 70%, yellow boxes in fig. 3) and
higher $NO_x$ concentrations (>150 pptv) $S_{NO}$ is negative, implying a net  NO sink of up to 6.4x$10^7$ molecules $cm^3$ $s^{-1}$
resembling the above mentioned PSS deviations in remote marine air masses (see fig. 6 and 7). However, during
humid conditions (RH > 70, blue boxes in fig. 3) $S_{NO}$ was positive with values of up to 5.1x$10^7$ molecules $cm^{-3}$ $s^{-1}$.
Due to low and invariant acetonitrile levels, anthropogenic activity and local biomass burning can be excluded as NO
source at this specific site. A net NO source during humid conditions is assumed to result from (biogenic) NO
emission from soil.  As shown in fig. 8, the PSS-based $S_{HONO}$ and $S_{NO}$ (time of day-averaged, excluding 3 days as
there are transition days 25.7. and 2.8. or the RH changed too quickly 15.7.) were highly correlated ($R^2$ = 0.78),
indicative of both reactive N-compounds being emitted from the same local source. Both HONO and NO have been
reported to be released from soil, with a strong dependency on soil water content (Su et al., 2011; Oswald et al.,
2013). The (dry state) soil humidification threshold level for NO emission is reported to be somewhat higher than for
HONO (Oswald et al., 2013), which might explain why a net PSS-based NO source was preferentially calculated for
higher relative humidity conditions, while for HONO the PSS indicated a daytime source under all humidity regimes
prevailing during the campaign. Analyzing microbial surface communities from drylands, Weber et al. (2015)
observed highly correlated NO-N and HONO-N emissions with Spearman rank correlation coefficients ranging
between 0.75 and 0.99. In this study, NO- and HONO-emissions were observed in drying soils with water contents
of 20-30% water holding capacity.
Even though we cannot make firm conclusions regarding the exact mechanism of HONO formation, the above
mentioned correlation analysis (and table 1) reveal that the instantaneous heterogeneous $NO_2$ conversion is not a
significant HONO source. We propose that HONO is emitted from nitrogen compounds being accumulated on
mountain slope soil surfaces produced either biologically by soil microbiota or from previously deposited $NO_y$. This



forms the major daytime HONO source responsible for morning concentration peaks and consistently high daytime
mixing ratios at the Cyprus field site. While biological formation is assumed to be more relevant for humid
conditions, physical $NO_y$ accumulation can be assumed to be stronger under dry conditions, as uptake coefficients
for a variety of trace gases were shown to be significantly higher for dry surfaces, among them $NO_2$ (Wang et al.,
2012, Liu et al., 2015), HONO (Donaldson et al. 2014a) and HCHO (Li et al., 2016). The strongest HONO morning
peaks observed after dry nights were accompanied by an increase in relative humidity driven by the sea breeze (fig.
4b), so we consider HONO being released preferentially under favourable humid conditions.
**5.4 OH production**
Many studies showed high contribution of HONO photolysis to the OH budget (up to 30% on daily average; Alicke
et al., 2002, Ren et al., 2006). Here the OH production rates is calculated based on the  main OH forming reactions,
which are the photolysis of ozone and subsequent reaction with water (R6+7), the photolysis of HONO (R2) and
HCHO (R8-11) and the reaction of alkenes with ozone (R12).
$$O_3 \xrightarrow{hv\ (<340\ nm)} O(^1D) + O_2 \qquad\qquad (R6)$$
$$O(^1D) + H_2O \rightarrow 2\ OH \qquad\qquad (R7)$$
$$HCHO \xrightarrow{hv\ (<370\ nm)} H + HCO\ (or\ H_2 + CO) \qquad\qquad (R8)$$
$$H + O_2 \rightarrow HO_2 \qquad\qquad (R9)$$
$$HCO + O_2 \rightarrow CO + HO_2 \qquad\qquad (R10)$$
$$HO_2 + NO\ or\ O_3 \rightarrow OH + NO_2\ or\ 2O_2 \qquad\qquad (R11)$$
$$alkene + O3 \rightarrow OH + other\ products \qquad\qquad (R12)$$
Reaction rates were taken from Atkinson et al. (2004) and Atkinson (1997). The water pressure over water was
calculated according to Murphy and Koop (2005). Reactions of $O(^1D)$ and $HO_2$ not forming OH are also considered.
OH formation yields of the reactions of alkenes with $O_3$ were taken from Paulson et al. (1999). Photolysis rates (J-
values) and concentrations of relevant compounds were as measured on Cyprus. Isoprene, α-pinene, β-pinene, Δ3-
carene and limonene were taken into account as the most relevant alkenes.
The results of this study are shown in fig. 9. All four production routes show a clear diel profile with higher
production rates during daytime. In the night only the reaction of alkene with $O_3$ produced significant amounts of OH
($1.5 \times 10^4$ molecules $cm^{-3}$ $s^{-1}$). With sunrise the other sources become more relevant. The photolysis of HONO and
HCHO lead to similar daytime OH production rates of about $0.8 - 1.7 \times 10^6$ molecules $cm^{-3}$ $s^{-1}$. The maximum OH
production rate by $O_3$ photolysis during daytime is about $1.5 \times 10^7$ molecules $cm^{-3}$ $s^{-1}$. In the morning and evening
hours the contribution of HONO photolysis to the total OH production is in average 30% (see fig. 9b) with peak
values of 60%, which is much higher than the contribution of $O_3$ photolysis at that time. During the rest of the day
the contribution decreases to 12%. The contribution of HCHO is slightly lower. At noon the most dominant OH
source is the photolysis of $O_3$.



## 6 Conclusion

Nitrous acid was found in low concentrations on the east Mediterranean Island of Cyprus during summer 2014. Daytime concentrations were much higher than during the night and about 30 times higher than would be expected by budget analysis based on photostationary state. The unknown source was calculated to be about $1.9 \times 10^6$ molecules $cm^{-3} s^{-1}$ around noon. Low $NO_x$ concentrations, high $HONO/NO_x$ ratio and low correlation between HONO and $NO_2$ indicate a local source which is independent from $NO_2$. Heterogeneous reactions of $NO_2$ on aerosols play an insignificant role during daytime. Emission from soil, either caused by photolysis of nitrate or gas-soil partitioning of accumulated nitrite/nitrous acid, is supposed to have a higher impact on the HONO concentration during this campaign. Also the NO budget analysis showed a missing source in the humid period, which correlates well with the unknown source of HONO, indicating a common source. The most likely source of HONO and NO is the emission from soil.

Even though the HONO concentration is only in the lower pptv level, it has a high contribution to the OH production in the early morning and evening hours.

**Acknowledgement**

This study was supported by the Max Planck Society (MPG) and the DFG-Research Center / Cluster of Excellence „The Ocean in the Earth System-MARUM".

We thank the Cyprus Institute and the Department of Labor Inspection for the logistical support, as well as the military staff at the Lara Naval Observatory in Ineia for the excellent collaboration.

Furthermore I´d like to thank Mathias Sörgel, an experienced colleague from MPI-C for his technical support on experimental set-up of atmospheric HONO measurements.

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

| | during the whole campaign | | | | | | | |
| | | | Time of day average | | | | | |
| | HONO | S$_{HONO}$ | HONO | S$_{HONO}$ | | | | |
|---|---|---|---|---|---|---|---|---|
| T | 0.006 | 0.135 | 0.488 | 0.227 | | | | |
| RH | 0.077 | 0.004* | 0.092 | 0.153 | | | | |
| Heat flux | 0.261 | 0.300 | *0.617$^c$* | *0.648$^c$* | | | | |
| J$_{NO2}$ | 0.263 | 0.414 | ***0.718$^b$*** | ***0.735$^b$*** | | | | |
| NO | 0.242 | 0.206 | **0.857$^a$** | *0.640$^c$* | | | | |
| NO$_2$ | 0.052 | 0.091 | 0.620$^c$ | 0.438 | | | | |
| NO$_2$*RH | 0.126 | 0.135 | 0.638$^c$ | 0.457 | | | | |
| NO$_2$*RH*aerosol surface | 0.095 | 0.110 | 0.256 | *0.561$^c$* | | | | |
| NO$_2$*J | 0.191 | 0.189 | **0.828$^a$** | **0.839$^a$** | | | | |
| NO$_2$*RH*J | 0.266 | 0.258 | **0.850$^a$** | **0.840$^a$** | | | | |
| NO$_2$*RH*J*aerosol surface | 0.221 | 0.218 | **0.806$^a$** | **0.848$^a$** | | | | |
| S$_{NO}$ | | 0.010 | | 0.268 | | | | |

**a** highly correlated R² > 0.8

**b** moderate correlated R² > 0.65

**c** poorly correlated R² > 0.5

***** anti-correlated

| | during the humid period | | | | during the dry period | | | |
| | | | Time of day average | | | | Time of day average | |
| | HONO | S$_{HONO}$ | HONO | S$_{HONO}$ | HONO | S$_{HONO}$ | HONO | S$_{HONO}$ |
|---|---|---|---|---|---|---|---|---|
| T | 0.006 | 0.126 | 0.031 | 0.113 | 0.120 | 0.013 | 0.453 | -0.015 |
| RH | 0.000 | 0.092* | 0.010* | 0.127* | 0.374 | 0.227 | ***0.730$^b$*** | ***0.683$^b$*** |
| Heat flux | 0.110 | 0.274 | 0.184 | *0.554$^c$* | *0.502$^c$* | 0.303 | ***0.685$^b$*** | *0.594$^c$* |
| J$_{NO2}$ | 0.150 | 0.467 | 0.245 | ***0.698$^b$*** | ***0.678$^b$*** | 0.357 | **0.829$^a$** | ***0.657$^b$*** |
| NO | 0.168 | 0.188 | 0.418 | ***0.676$^b$*** | 0.487 | 0.323 | ***0.730$^b$*** | 0.302 |
| NO$_2$ | 0.066 | 0.075 | 0.300 | 0.353 | 0.037 | -0.002* | *0.619$^c$* | 0.171 |
| NO$_2$*RH | 0.084 | 0.053 | 0.294 | 0.245 | 0.161 | 0.021 | ***0.714$^b$*** | *0.523$^c$* |
| NO$_2$*RH*aerosol surface | 0.047 | 0.079 | 0.111 | 0.147 | 0.241 | 0.106 | *0.557$^c$* | *0.621$^c$* |
| NO$_2$*J | 0.214 | 0.291 | 0.427 | **0.910$^a$** | 0.358 | 0.018 | **0.872$^a$** | ***0.657$^b$*** |
| NO$_2$*RH*J | 0.231 | 0.271 | 0.467 | **0.850$^a$** | 0.434 | 0.085 | **0.820$^a$** | ***0.770$^b$*** |
| NO$_2$*RH*J*aerosol surface | 0.140 | 0.160 | 0.465 | *0.784$^b$* | 0.414 | 0.171 | 0.664$^b$ | *0.678$^b$* |
| S$_{NO}$ | | 0.323 | | *0.778$^b$* | | 0.003* | | -0.009* |




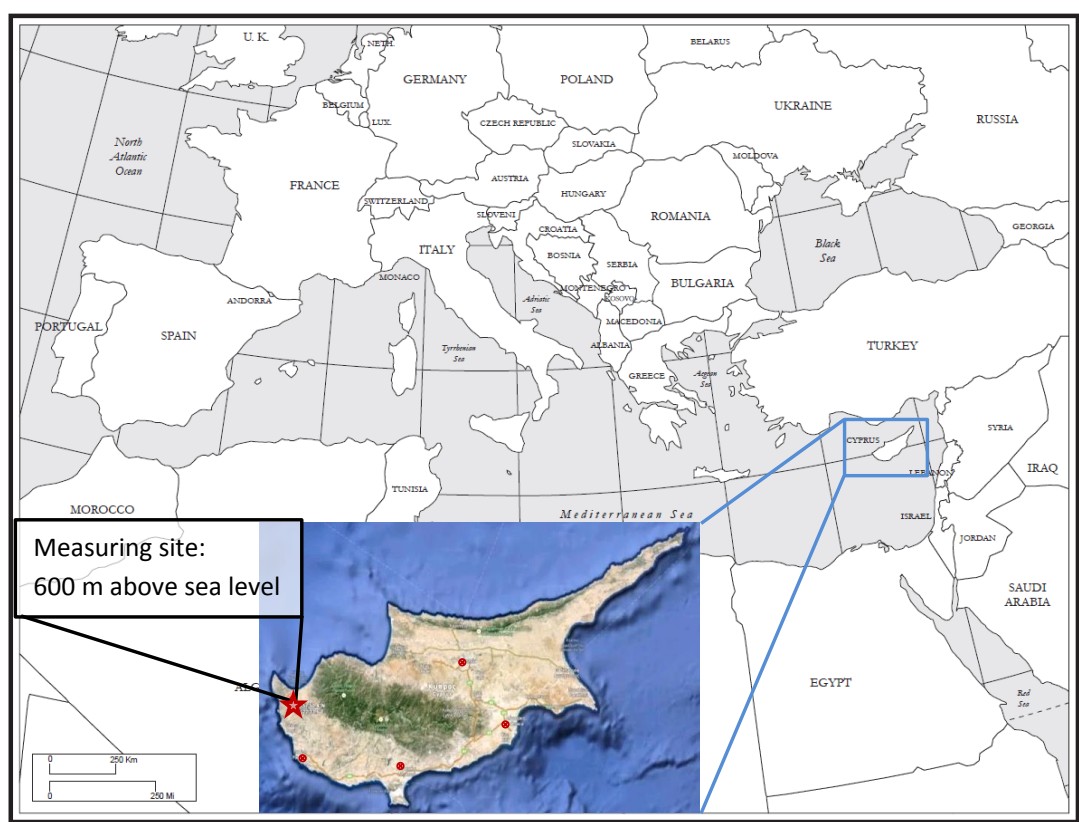

**Figure 1: Map of location: the red star shows the location of Ineia and the measuring site. The four red points mark the main cities of Cyprus, Nicosia, Larnaca, Limassol and Paphos (clockwise ordering), map produced by the Cartographic Research Lab University of Alabama, map of Cyprus: google maps.**



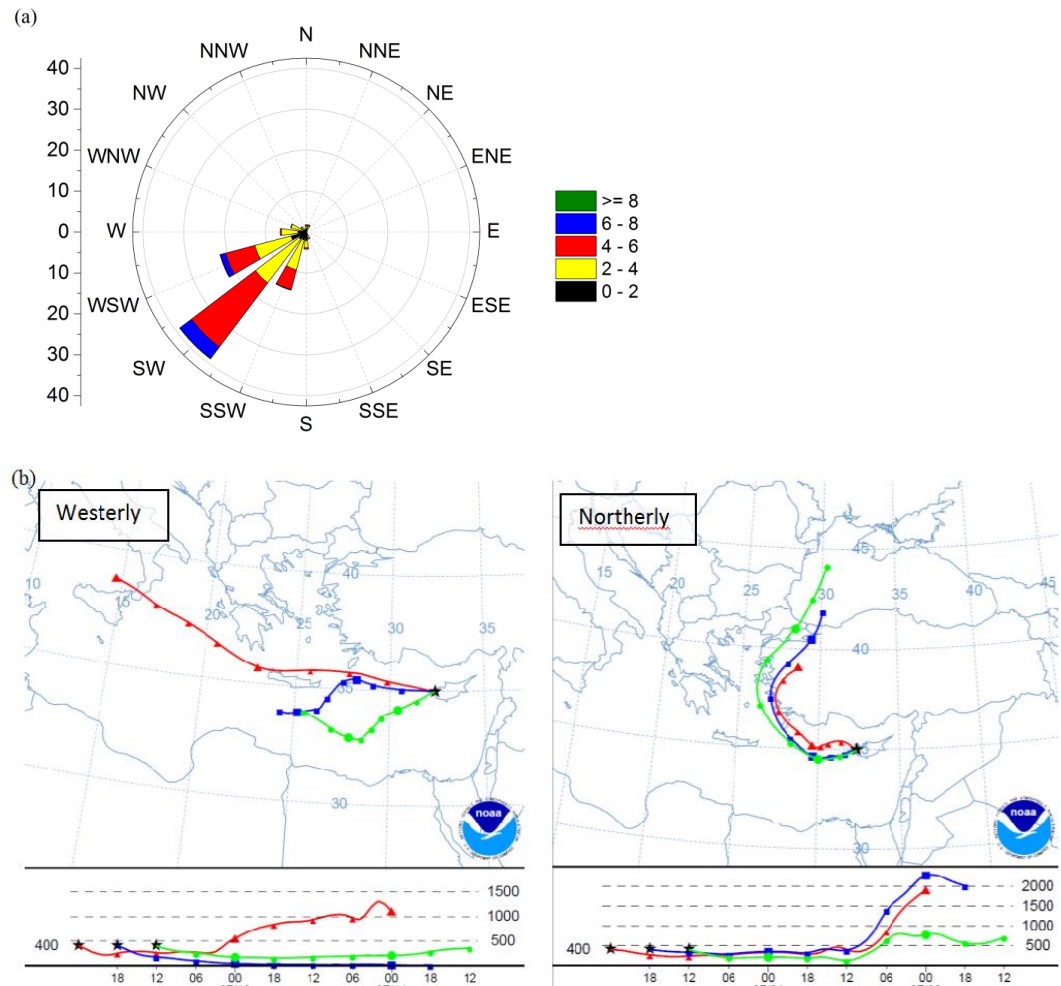

**Figure 2: Airflow conditions during the CYPHEX campaign: a) Measured local wind direction, b) back trajectories calculated with NOAA Hysplit model showing examples for the two main air mass origins (48 hours, UTC = LT - 3 h).**





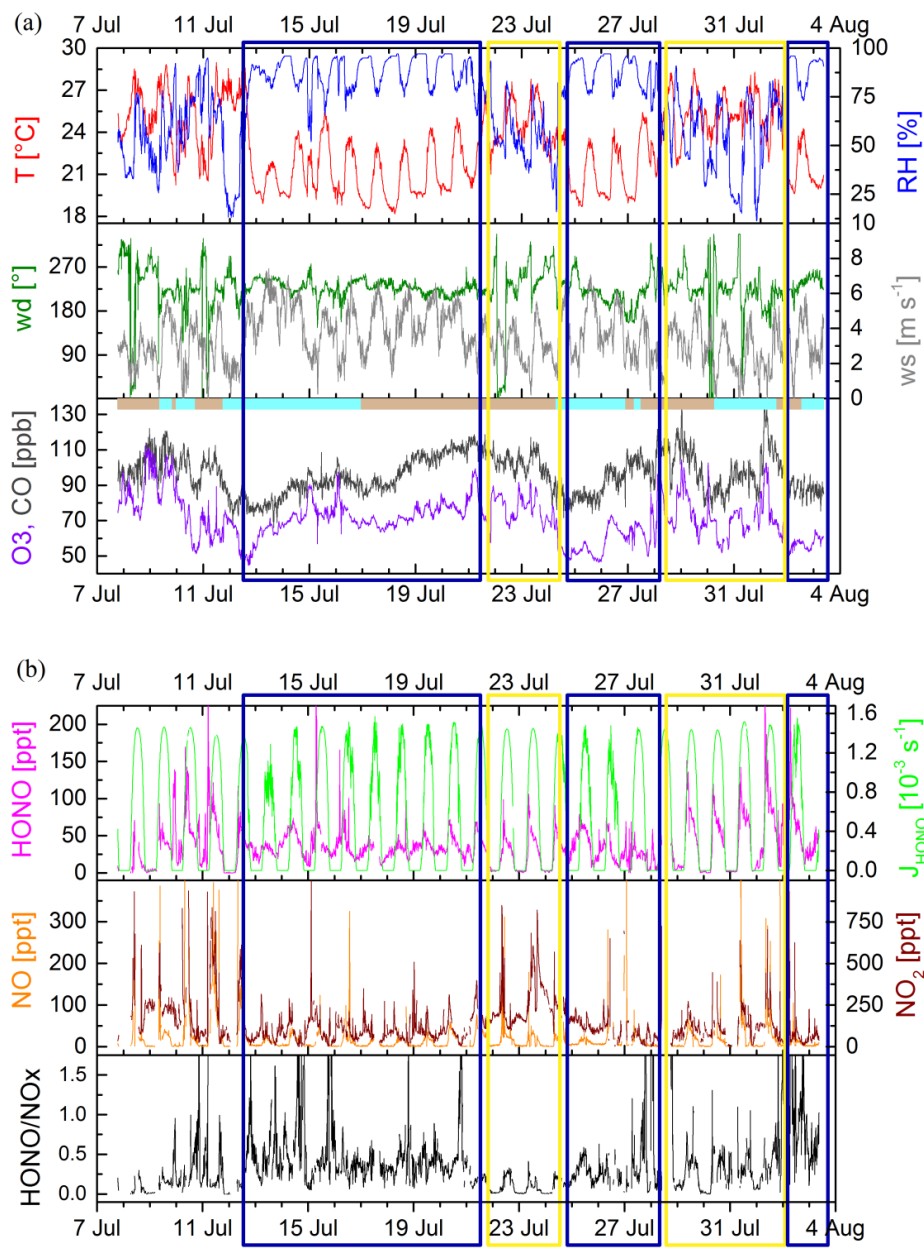

**Figure 3: Measured variables during the whole campaign from 7th July to 4th August 2014, a) meteorological data (Temperature T, relative humidity RH, wind direction and speed wd, ws) and O3 and CO indicating stable conditions, in the lower panel the bar indicates the air mass origin: bright blue = westerly, brownish = northerly, b) observed mixing ratios of HONO, NO2 and NO, and the photolysis frequency JHONO and the HONO/NOx ratio. The yellow and blue boxes reflect the dry and the humid periods, respectively.**



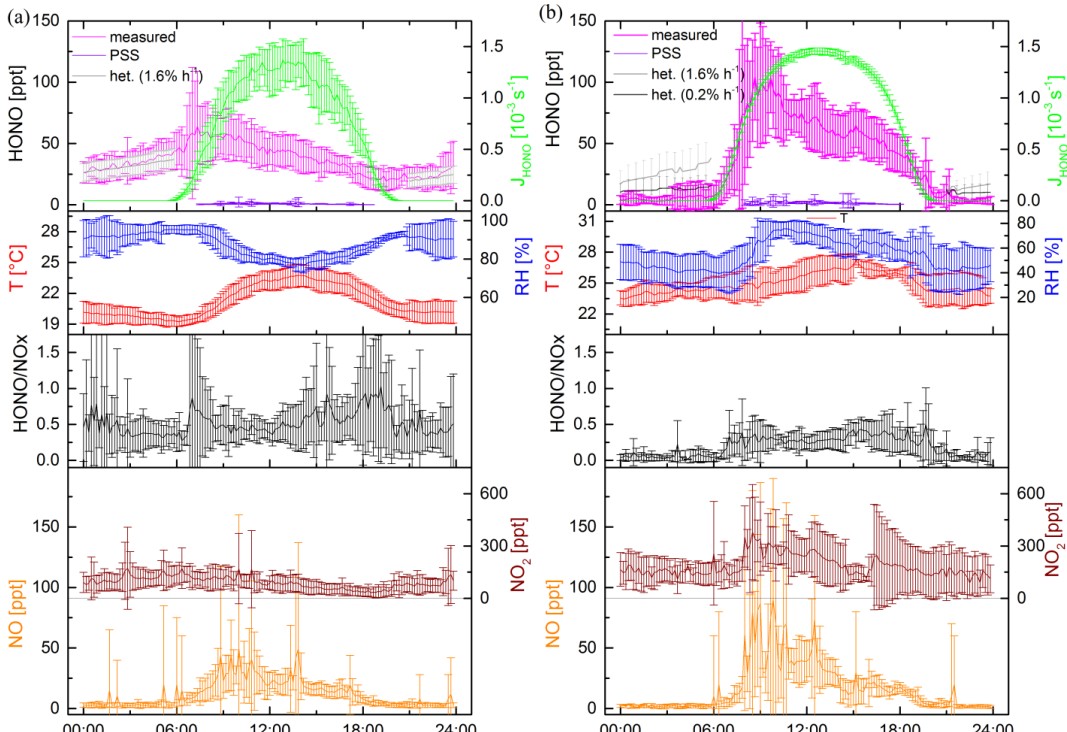

**Figure 4: Diel variation of meteorological data (Temperature T, relative humidity RH), NO and NO₂ mixing ratios, the photolysis rate for HONO J_HONO and HONO mixing ratios (pink: measured, violet: daytime photostationary state PSS, grey: nighttime heterogeneous NO₂ conversion) and HONO/NOx ratio for a) average for period when RH was above 60% (blue box in Fig. 3) and b) average for dry period when RH was below 60% (yellow box in Fig. 3).**

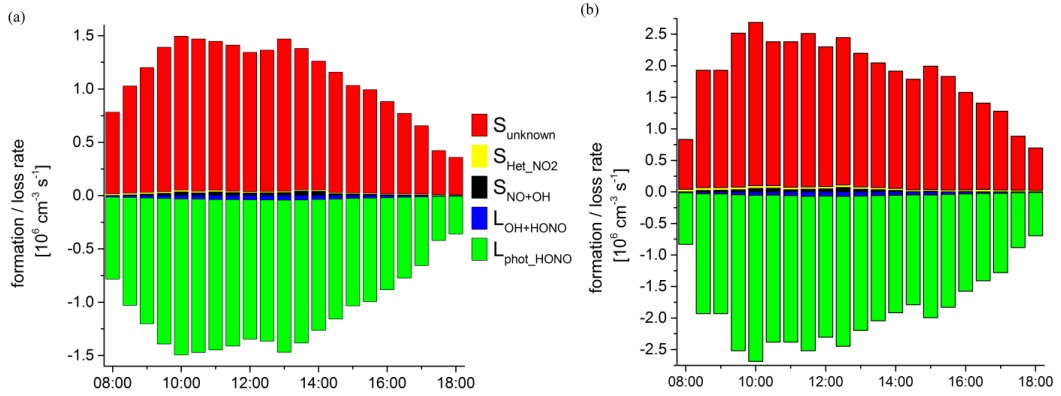

**Figure 5: HONO budget analysis for a) the humid and b) the dry period. $S_{OH+NO}$ (black) stands for the formation rate of HONO via the reaction of NO and OH, $S_{Het\_NO2}$ (yellow) is the formation rate for the heterogeneous reaction of NO₂ (conversion rate 1.6% h⁻¹), $L_{phot}$ (green) and $L_{OH+HONO}$ (blue) are the loss rates via photolysis and the reaction with OH and $S_{unknown}$ is the unknown source.**





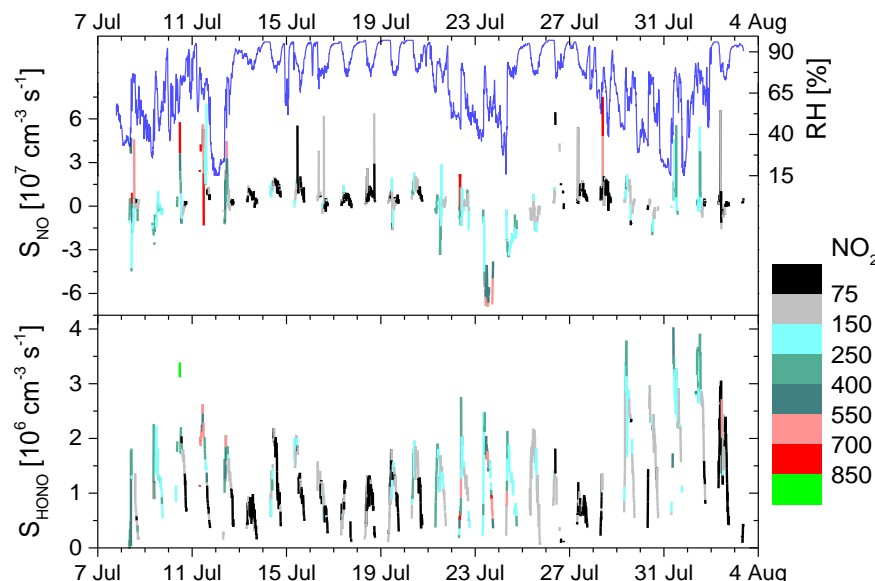

**Figure 6:** NO$_2$ (color-coded) and RH dependence of the sources of NO (S$_{NO}$) and HONO (S$_{HONO}$).

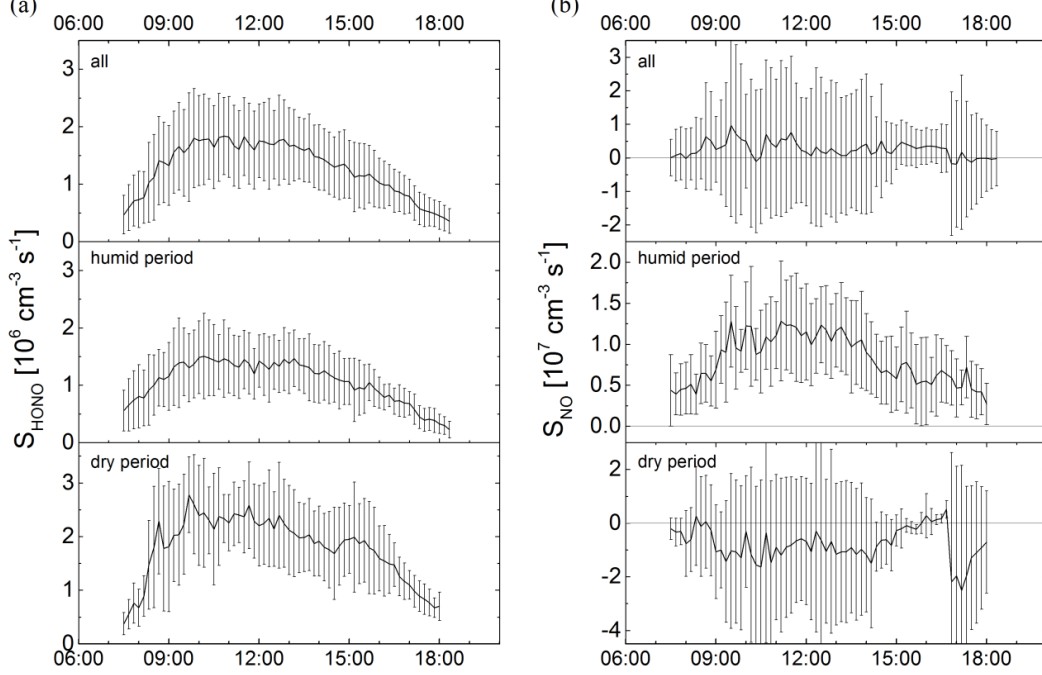

**Figure. 7:** Diel profile of both unknown sources S$_{HONO}$ (a) and S$_{NO}$ (b) for all data, humid (excluding transition days: 25.7. and 2.8 and 15.7. as RH conditions changed too quickly) and dry periods.





**Figure 8: Correlation of $S_{HONO}$ to light induced $NO_2$ reaction (for both periods, humid = blue triangle, dry = orange square), to NO and $S_{NO}$ (only for humid period); time of day average data were used ($S_{HONO}$ and NO; $S_{HONO}$ and $S_{NO}$ excluding the 3 days mentioned before).**





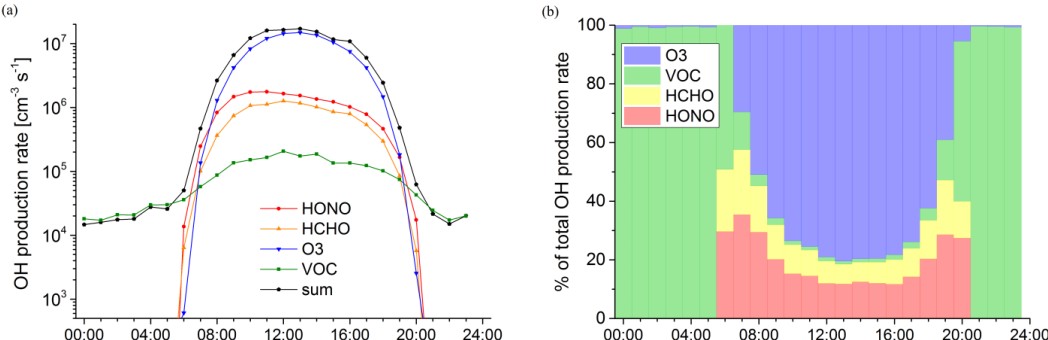

**Figure. 9: Average diel pattern of OH production from HONO, O$_3$, HCHO and VOC, a) shown as production rate and b) percentage contributions to total OH production.**