# Peer review of "Daytime formation of nitrous acid at a coastal remote site in"

_Atmospheric Chemistry and Physics, 2016_

## Referee Comment (RC1) · Anonymous Referee #1 · 22 Aug 2016

**Review: Daytime formation of nitrous acid at a coastal remote site in Cyprus indicating a common ground source of atmospheric HONO and NO, by Meusel et al.**

General comments

In this manuscript the authors present results of HONO and other trace gas species from a study performed in Cyprus as part of the CYPHEX campaign in 2014. During the measurement period they observed a high HONO/NOx ratio and a large daytime source of HONO. A budget analysis is performed and a missing source of HONO up to 3.4 x 10$^6$ molecules cm$^{-3}$ s$^{-1}$ calculated, which is comparable to values reported in mountain and forest sites. Under humid conditions the HONO source correlates well with NO and the authors attribute this missing HONO source to emissions from soil. Finally, the impact of the HONO on OH production rates is calculated and the results show that the HONO photolysis contributes, on average, 30% to OH production during the morning and evening. Understanding the daytime source of HONO is important due to its role in OH formation and this study provides important data on HONO sources in a location which is not strongly impacted by combustion sources.

The manuscript is well written, with appropriate sections and easy to follow. I recommend the manuscript for publication in ACP after addressing the comments below:

Specific comments

One of the main concerns is that no uncertainty analysis has been performed for the HONO/NOx ratios or the HONO budget and calculation of the missing HONO sources. This should include instrument uncertainties in the HONO and NOx measurements along with errors in the PSS calculation. It would then be beneficial to include error bars on Figure 5a and b, to show the upper and lower limits to the estimated unknown HONO source.

In section 5.1, the heterogeneous reaction of NO2 to form HONO is estimated by applying an NO$_2$-HONO conversion rate of 1.6% h$^{-1}$ overnight. Under humid conditions the estimated values agree well with measured values. A much lower rate of 0.22 % h$^{-1}$ was applied in the drier period, which the author's state matches better to their observations. However, Fig 4. shows the measured HONO is still lower than the estimated values during some periods overnight. Perhaps it would be better to determine a conversion rate under dry conditions for this site using the NOx scaling approach (e.g. Sorgel et al., 2011) to compare with other studies, as I expect it is lower.

Pg 3, L25-26. Please state the uncertainty of the HONO measurements here too.

Pg 6, L18. The ± values in the parenthesis should be clarified. Are these 1-sigma standard deviation of the mean?

Pg 7, L7. It is stated that the mean NO mixing ratios are close to the detection limit at 2 pptv, however, this is actually below the detection limit, which is given as 5 pptv on Pg 4, L13.

Pg 8, L5-7. Here, HONO mixing ratios are estimated and compared to the measured HONO overnight using a conversion factor between $NO_2$ and HONO of 1.6% $h^{-1}$. The authors cite three studies where this value has been determined, although, it should be made clear here that a range of values were reported across these studies.

Pg 9, L25. Please state the values for $k_1$ and $k_2$ used in Eq. 2.

Fig 4: The error bars in figure 4b for the 0.2% rate are difficult to see, please use a darker color or use thicker lines.

In Figure 5, the caption states that a conversion rate of 1.6% $h^{-1}$ is used for $S_{Het\_NO2}$, however, Figure 4b shows that a lower rate (0.22% $h^{-1}$) is more appropriate for the dry period. Please clarify which rate you use for Fig 5b.

Fig 6. Include units for $NO_2$ in the legend.

Fig. 4 and Fig 7. Please state in the figure captions what the error bars represent.

References

Sörgel, M., Trebs, I., Serafimovich, A., Moravek, A., Held, A., and Zetzsch, C.: Simultaneous HONO measurements in and above a forest canopy: influence of turbulent exchange on mixing ratio differences, Atmos. Chem. Phys., 11, 841-855, doi:10.5194/acp-11-841-2011, 2011.

---

## Referee Comment (RC2) · Anonymous Referee #2 · 25 Aug 2016

This paper uses measurements of HONO with a wide range of supporting data to assess sources of HONO in the remote coastal site in Cyprus. The findings are that there is a common source of HONO and NO and it is speculated that this is emission from microbial communities on soil surfaces. The work is important as HONO provides a route to OH radicals that is often not considered and sources of HONO in both urban and remote regions are uncertain. The authors have done a good job presenting their data and the conclusions they draw are reasonable. It undoubtedly adds to the sphere of knowledge surrounding atmospheric HONO. The paper is well presented with good clear figures and should be published in ACP subject to the authors addressing the

following comments.

General comments:

The main conclusion of the paper is that there is a soil source of HONO and NO, which is arrived upon by looking at correlations between the 'missing' HONO source (i.e. the difference between HONO calculated using a steady state approximation including a series of known sources and the measured HONO) and a missing source of NO (based on NO deviations from the Leighton ratio). A strong correlation is given as evidence of a common source. Is this source thought to be photolytically driven? If not why are observations of NO at night seemingly zero (although it is quite difficult to see the exact levels on the plots), where as HONO is shown to increase during the night. Maybe this is just a result of NO reacting with O3 before the measurement location but the authors should clarify this. How far from the potential soil emission source in the measurement site? The authors should also comment on how this effects the validity of the steady state approximation, with reference to the Lee et al. 2003 study that gives caveats for the use of a steady state approximation to interpret HONO measurements.

I find the analysis of OH production showing the importance of HONO confusing because it details production of OH from HCHO, which is indirect and requires conversion of the HO2 produced with NO to form OH. I believe it would be better to just include HO2 + NO as an OH source, regardless of where the HO2 is coming from. Another option would be to have a total HOx radical budget analysis. The authors should also comment on the fact that the HONO source here is only important near to the surface (an estimate could be made of the vertical structure of HONO) and thus it is not relevant for the entire troposphere. This is important when considering HONO as an atmospheric 'oxidant'.

The authors mention in the experimental description that OH was measured during the field campaign but there is then no further mention of it in the manuscript. Have the authors (or anyone else) examined the OH data to assess if the measured HONO is

required to close the HOx budget? I realise this may be the subject of further publications but if it is stated that HOx was measured it seems odd that no mention is made of the results.

The manuscript is generally well referenced however a recent study by Lee et al. 2016 in London contains a lot of detail about potential HONO sources in an urban area and should be referenced. There is also a recent study by Mamtimin et al (2016) which discusses biogenic NO and HONO emissions that seems to be extremely relevant to this work. The authors should comment on hwo their results compare to this.

Minor comments:

The authors should make sure they clarify what the error bars on plots and in the text actually refer to (e.g. figures 4 and 7) P. 12 line 11: Use O3 rather than ozone as has been done in the rest of the manuscript P. 7 line 7: there is a discrepancy between the detection limit stated here (2pptv) and that in the experimental section (5pptv) – please confirm.

References:

Mamtimin, B., Meixner, F. X., Behrendt, T., Badawy, M., and Wagner, T.: The contribution of soil biogenic NO and HONO emissions from a managed hyperarid ecosystem to the regional NOx emissions during growing season, Atmos. Chem. Phys., 16, 10175-10194, doi:10.5194/acp-16-10175-2016, 2016.

Lee, J. D., Whalley, L. K., Heard, D. E., Stone, D., Dunmore, R. E., Hamilton, J. F., Young, D. E., Allan, J. D., Laufs, S., and Kleffmann, J.: Detailed budget analysis of HONO in central London reveals a missing daytime source, Atmos. Chem. Phys., 16, 2747-2764, doi:10.5194/acp-16-2747-2016, 2016.

Lee, B. H., Wood, E. C., Herndon, S. C., Lefer, B. L., Luke, W. T., Brune, W. H., Nelson, D. D., Zahniser, M. S., and Munger, J. W.: Urban measurements of atmospheric nitrous acid: A caveat on the interpretation of the HONO photostationary state, J. Geophys.

Res., vol. 118, 12274–12281, doi:10.1002/2013JD020341, 2013.

---

## Author Comment (AC2) · 27 Oct 2016

**General comments:**

*This paper uses measurements of HONO with a wide range of supporting data to assess sources of HONO in the remote coastal site in Cyprus. The findings are that there is a common source of HONO and NO and it is speculated that this is emission from microbial communities on soil surfaces. The work is important as HONO provides a route to OH radicals that is often not considered and sources of HONO in both urban and remote regions are uncertain. The authors have done a good job presenting their data and the conclusions they draw are reasonable. It undoubtedly adds to the sphere of knowledge surrounding atmospheric HONO. The paper is well presented with good clear figures and should be published in ACP subject to the authors addressing the following comments.*

> **Response:**
> We thank the reviewer for the positive feedback. Please find our point-to-point responses listed below.

**Specific comment:**

*The main conclusion of the paper is that there is a soil source of HONO and NO, which is arrived upon by looking at correlations between the 'missing' HONO source (i.e. the difference between HONO calculated using a steady state approximation including a series of known sources and the measured HONO) and a missing source of NO (based on NO deviations from the Leighton ratio). A strong correlation is given as evidence of a common source. Is this source thought to be photolytically driven? If not why are observations of NO at night seemingly zero (although it is quite difficult to see the exact levels on the plots), whereas HONO is shown to increase during the night. Maybe this is just a result of NO reacting with $O_3$ before the measurement location but the authors should clarify this.*

> **Response:**
> True, the correlation analysis was based on daytime values only. For nighttime conditions, the chemistry is relatively slow and transport processes could strongly influence the budget of nitrogen-containing species, that's why we focus on the daytime chemistry.
> The difference in nighttime accumulation of NO and HONO may be due to other reasons, like $NO_2$ heterogeneous conversion, being relevant for HONO accumulation within a shallow nocturnal boundary layer (here 0.4-1.6 %, in line with other literature, see chapter "5.1 nighttime HONO accumulation"), while there is no chemical source for NO. Also the nighttime reaction of NO + OH forming HONO would result in a preference in HONO accumulation, with nighttime OH concentration sometimes as high as $1 \times 10^6$ molecules $cm^{-3}$ (see Fig. S3). As suggested by the referee, NO titration by the reaction with $O_3$ may also play a role for the absence of nighttime NO accumulation, with continuously high $O_3$ concentrations (60-90 ppb), while there is no major loss of HONO due to the lack of photolysis. Another option would be different temperature dependencies of NO and HONO emissions from soil (e.g., Oikawa et al., 2015; Mamtimin et al., 2016, which is now stated on page 12 line 31-35).
> We modified one sentence of the "result" chapter, page 7, line 22). "In the absence of local NO sources low nighttime values are a result of the conversion of NO to $NO_2$ by $O_3$ which was continuously high (Beygi et al., 2011)"

**Specific comment:**

*How far from the potential soil emission source in the measurement site? The authors should also comment on how this effects the validity of the steady state approximation, with reference to the Lee et al. 2013 study that gives caveats for the use of a steady state approximation to interpret HONO measurements.*

**Response:**

Thanks to the comment. We are afraid that we may not have described our calculation properly and the using of HONO$_{PPS}$ was misleading. We actually followed the method in Su et al. (2008a) to calculate HONO missing source. With this method, we did not assume HONO to be at PSS, because the measured d[HONO]/dt has been accounted in the HONO missing source estimation and according to our measurement d[HONO]/dt was not equal to zero, which did mean that HONO was not at PPS.

$$S_{HONO} = J_{HONO}[HONO] + k_2[OH][HONO] - k_1[OH][NO] - k_{het}[NO_2] + \frac{\Delta[HONO]}{\Delta t}$$

Lee et al. (2013) states that assuming HONO to be completely at PSS will likely overestimate the strength of any "unknown source". In the Lee et al. (2013) case, the authors would come up with up to 1.1 ppb h$^{-1}$ with the PSS approach. They argue that instead of presuming a PSS, they can explain the observed HONO from pure precursor chemistry by applying a simple chemical box model.

Lee et al. (2013) argue that the PSS assumption might not have been valid for their case study, because the transport time from nearby NO$_x$ vehicle exhaust emission sources to the measurement site was likely less than the time required for HONO to reach PSS. Using a chemical box model, Lee et al. (2013) demonstrated that there is initially net HONO formation from assumed strong emissions (100 ppm for the sum of NO, NO$_2$, HONO), as high levels of NO in vehicle exhaust react with assumed entrainment of ambient OH. In the respective model daytime show-case, this HONO net production (d[HONO]/dt>0) , sustained for 2.5 min after precursor emission. Subsequently, net HONO loss dominated by photolysis led to a negative d[HONO]/dt in their calculations, which was sustained for several minutes until PSS is established after ca. 10 min (for mean daytime conditions) or up to several hours depending on time of day. This way, Lee et al. (2013) claim that "for all conditions, d[HONO]/dt is negative for a specific period of time, during which sampling vehicle exhaust can lead to overestimates of secondary HONO sources *if a photostationary state is inappropriately assumed*", and hence "… there exists a window of time in which d[HONO]/dt is negative. Erroneously assuming the presence of PSS during this time period would lead to overestimates of secondary HONO sources."

With respect to our analysis, first of all, we did not assume that HONO PSS is fully established at our measurement site (d[HONO]/dt was not equal to 0). However, even though for the Cyprus case, the mean upwind distance between the measurement site and the coast line is about 6 km. With a mean wind velocity of 3 m s$^{-1}$ the respective air mass travel time over land/soil surface is about half an hour, i.e., several times the daytime lifetime of HONO. Moreover, and in strong contrast to Lee et al. (2013), at the Cyprus site the concentrations of HONO precursors were extremely low. In the Cyprus case, the observed atmospheric load of precursors (NO and OH) is by far too low to explain the observed HONO concentrations, or d[HONO]/dt, respectively (see Fig. 5). Even doubling the contribution of the chemical source (NO + OH) would not lead to a substantial reduction of the strength of the calculated un-identified HONO source.

To account for any caveats of any PSS assumptions, we now state in the text (page 10, line 14-30):

"Lee et al. (2013) argue that the HONO PSS assumption might overestimate the strength of any un-identified source, if the transport time from nearby NOx emission sources to the measurement site is less than the time required for HONO to reach PSS. In this study, the missing source was calculated according to Su et al., 2008a (eq.3), where PSS was not assumed. Also in our measurements, dHONO/dt was not equal to zero, as HONO was not at PSS.

$$S_{HONO} = J_{HONO}[HONO] + k_2[OH][HONO] - k_1[OH][NO] - k_{het}[NO_2] + \frac{\Delta[HONO]}{\Delta t} \qquad \text{(Eq.3)}$$

with [HONO] being the measured HONO concentration and $k_{het}$ the heterogeneous conversion rate of $NO_2$ to HONO, which was discussed above to be 1.6% $h^{-1}$ during the wet period and 0.36% $h^{-1}$ during the dry period. $\Delta[HONO]/\Delta t$ is the observed change of HONO concentration unequal to 0. The uncertainty of the calculated missing source $S_{HONO}$ was estimated to be about 16% based on the Gaussian error propagation of instrument uncertainties of HONO, NO, $NO_2$, J and OH.

Nevertheless, at the study site of Cyprus, the mean upwind distance between the measurement site and the coast line was about 6 km, and the mean wind velocity was about 3 m $s^{-1}$. Accordingly, the respective air mass travel time over land is estimated to be about half an hour, which is somewhat longer than the daytime lifetime of HONO and might provide enough time for the equilibrium processes. Furthermore and in a strong contrast to Lee et al. (2013), at the Cyprus site the concentrations of HONO precursors (NO and OH) were extremely low, by far too low to explain the observed HONO concentrations."

**Specific Comment:**
*I find the analysis of OH production showing the importance of HONO confusing because it details production of OH from HCHO, which is indirect and requires conversion of the HO$_2$ produced with NO to form OH. I believe it would be better to just include HO$_2$ + NO as an OH source, regardless of where the HO$_2$ is coming from. Another option would be to have a total HOx radical budget analysis.*

**Response:**
We thank the referee for disclosing this critical detail. OH budget analysis including photolysis of HCHO was done before e.g. by Alicke et al., 2002 but then also $RO_2$ primary production should be considered as it will also be converted in OH through cycling processes. HONO photolysis, ozonolysis of alkenes and photolysis of $O_3$ and subsequent reaction with water contribute to the primary OH production. HCHO photolysis firstly forms $HO_2$ which is in fast equilibrium with OH e.g. via the reaction of $HO_2$ and NO. Therefore it contributes to secondary OH production. In this study we focus on the evaluation of HONO sources and wanted to give a brief outlook on its importance on OH. To realize this we just show the primary OH production routes (and deleted the OH production via HCHO photolysis). Furthermore we changed the term "OH production" into "primary OH production".
As written below, a complete detailed HOx budget analysis will be published from colleagues soon.

[Figure]

Revised fig. 9: Average diel pattern of primary OH production from HONO, $O_3$ and VOC, shown as a) production rate and b) percentage contributions to primary OH production.

**Specific Comment:**

*The authors should also comment on the fact that the HONO source here is only important near to the surface (an estimate could be made of the vertical structure of HONO) and thus it is not relevant for the entire troposphere. This is important when considering HONO as an atmospheric 'oxidant'.*

**Response:**

Indeed, many studies have shown decreasing HONO mixing ratios with altitude in the lowest few hundred meters of the troposphere (Vogel et al., 2003; Zhang et al., 2009; Young et al., 2012; Wong et al., 2012 and 2013; VandenBoer et al., 2013). According to the modelling results of Wong et al. 2013, we estimate that the ground HONO source could be important for up to 200–300 m a.g.l. According to the referee's suggestion we now state in the introduction (Page 3, line 9-13):

"Many studies have shown decreasing HONO mixing ratios with altitude in the lowest few hundred meters of the troposphere, due to respective short atmospheric lifetime compared to vertical transport time (Wong et al., 2012 and 2013; Vogel et al., 2003; VandenBoer et al., 2013; Zhang et al., 2009; Young et al., 2012; ). According to the modelling results of Wong et al. 2013, we estimate that the ground HONO source could be important for up to 200–300 m a.g.l. This indicates that HONO is more relevant for the OH budget close to the surface than in high altitude air masses."

**Comment:**

*The authors mention in the experimental description that OH was measured during the field campaign but there is then no further mention of it in the manuscript. Have the authors (or anyone else) examined the OH data to assess if the measured HONO is required to close the HOx budget? I realise this may be the subject of further publications but if it is stated that HOx was measured it seems odd that no mention is made of the results.*

**Response:**

We thank the referee for this suggestion. In this study, we focused on HONO and its missing daytime source. OH data were used to calculate the HONO budget and $HO_2$ data were used to study the NO budget. The potential contribution of HONO photolysis to OH production is studied in a short chapter showing different OH production routes. A "total $HO_x$ radical budget analysis" is not focus of this manuscript. A future CYPHEX paper of colleagues will deal with the $HO_x$ budget closure study,

including a detailed box modelling approach for the total HO$_x$ budget and OH recycling. The manuscript will be submitted soon.

**Comment:**

*The manuscript is generally well referenced however a recent study by Lee et al. 2016 in London contains a lot of detail about potential HONO sources in an urban area and should be referenced. There is also a recent study by Mamtimin et al. (2016) which discusses biogenic NO and HONO emissions that seems to be extremely relevant to this work. The authors should comment on how their results compare to this.*

**Response:**

We greatly appreciate these reference suggestions. Both fit well into this study.

Mamtimin et al., 2016 is now cited in the introduction (page 3, line3), and twice in "common daytime source of HONO and NO" (page 12, line 28 and line 31-35) when comparing with other studies on HONO and NO emission.
"Mamtimin et al. (2016) investigated HONO and NO emissions of natural desert soil and with grapes or cotton cultivated soils in an oasis in the Taklamakan desert in the Xinjiang region in China. After irrigation they didn´t find direct emission, but when the soil had almost dried out (gravimetric soil water content 0.01-0.3) emissions up to 115 ng N m$^{-2}$ s$^{-1}$ were detected. In addition they observed soil-temperature dependent emission of reactive nitrogen.

Lee et al. (2016) is now cited once in the introduction (page 3, line 17) and in "daytime HONO budget" (page 11, line 15-16), discussing possible light induced HONO formation and comparing correlation factors:
"Lee et al. (2016) found even lower correlation with [NO$_2$] (R$^2$ = 0.0001) but similar good correlation with J$_{NO2}$*[NO$_2$] (R$^2$=0.70) at an urban background site in London."

**Minor comments:**

*The authors should make sure they clarify what the error bars on plots and in the text actually refer to (e.g. figures 4 and 7)*

Correct. As also suggested by Referee #1, we now clarify what the error bars on plots and in the text actually refer to. Error bars in Fig. 4 and 7 (and ± values) indicate standard deviation (1 sigma).

*P. 12 line 11: Use O$_3$ rather than ozone as has been done in the rest of the manuscript*

Thanks for noticing. Is now changed accordingly in the text.

*P. 7 line 7: there is a discrepancy between the detection limit stated here (2pptv) and that in the experimental section (5pptv) – please confirm.*

Thanks for indicating. As also suggested by referee #1, a detection limit of 5 pptv is now stated in the revised versions of the manuscript.

References:

Mamtimin, B., Meixner, F. X., Behrendt, T., Badawy, M., and Wagner, T.: The contribution of soil biogenic NO and HONO emissions from a managed hyperarid ecosystem to the regional NOx emissions during growing season, Atmos. Chem. Phys., 16, 10175-10194, 2016.

Lee, J. D., Whalley, L. K., Heard, D. E., Stone, D., Dunmore, R. E., Hamilton, J. F., Young, D. E., Allan, J. D., Laufs, S., and Kleffmann, J.: Detailed budget analysis of HONO in central London reveals a missing daytime source, Atmos. Chem. Phys., 16, 2747-2764, 2016.

Lee, B. H., Wood, E. C., Herndon, S. C., Lefer, B. L., Luke, W. T., Brune, W. H., Nelson, D. D., Zahniser, M. S., and Munger, J. W.: Urban measurements of atmospheric nitrous acid: A caveat on the interpretation of the HONO photostationary state, J. Geophys. Res., vol. 118, 12274–12281, 2013.

Oikawa, P. Y., Ge, C., Wang, J., Eberwein, J. R., Liang, L. L., Allsman, L. A., Grantz, D. A., and Jenerette, G. D.: Unusually high soil nitrogen oxide emissions influence air quality in a high-temperature agricultural region, Nat. Commun., 6, 2015.

Alicke, B., Platt, U., Stutz, J.: Impact of nitrous acid photolysis on the total hydroxyl radical budget during the limitation of oxidant production/pianura padana produzione di ozono study in Milan. Journal of Geophysical Research 107 (D22), 8196, 2002.

Vogel, B., Vogel, H., Kleffmann, J., and Kurtenbach, R.: Measured and simulated vertical profiles of nitrous acid - Part II. Model simulations and indications for a photolytic source, Atmospheric Environment, 37, 2957-2966, 2003.

VandenBoer, T. C., Brown, S. S., Murphy, J. G., Keene, W. C., Young, C. J., Pszenny, A. A. P., Kim, S., Warneke, C., de Gouw, J. A., Maben, J. R., Wagner, N. L., Riedel, T. P., Thornton, J. A., Wolfe, D. E., Dubé, W. P., Öztürk, F., Brock, C. A., Grossberg, N., Lefer, B., Lerner, B., Middlebrook, A. M., and Roberts, J. M.: Understanding the role of the ground surface in HONO vertical structure: High resolution vertical profiles during NACHTT-11, Journal of Geophysical Research: Atmospheres, 118, 10,155-110,171, 2013.

Wong, K. W., Tsai, C., Lefer, B., Haman, C., Grossberg, N., Brune, W. H., Ren, X., Luke, W., and Stutz, J.: Daytime HONO vertical gradients during SHARP 2009 in Houston, TX, Atmospheric Chemistry and Physics, 12, 635-652, 2012.

Wong, K. W., Tsai, C., Lefer, B., Grossberg, N., and Stutz, J.: Modeling of daytime HONO vertical gradients during SHARP 2009, Atmospheric Chemistry and Physics, 13, 3587-3601, 2013.

Young, C. J., Washenfelder, R. A., Roberts, J. M., Mielke, L. H., Osthoff, H. D., Tsai, C., Pikelnaya, O., Stutz, J., Veres, P. R., Cochran, A. K., VandenBoer, T. C., Flynn, J., Grossberg, N., Haman, C. L., Lefer, B., Stark, H., Graus, M., de Gouw, J., Gilman, J. B., Kuster, W. C., and Brown, S. S.: Vertically Resolved Measurements of Nighttime Radical Reservoirs; in Los Angeles and Their Contribution to the Urban Radical Budget, Environmental Science & Technology, 46, 10965-10973, 2012.

Zhang, N., Zhou, X. L., Shepson, P. B., Gao, H. L., Alaghmand, M., and Stirm, B.: Aircraft measurement of HONO vertical profiles over a forested region, Geophysical Research Letters, 36, 2009.

---

## Author Response (AR1)

**Anonymous Referee #1**

**General comments:**
*In this manuscript the authors present results of HONO and other trace gas species from a study performed in*
*Cyprus as part of the CYPHEX campaign in 2014. During the measurement period they observed a high HONO/NOx*
*ratio and a large daytime source of HONO. A budget analysis is performed and a missing source of HONO up to 3.4*
*x $10^6$ molecules $cm^{-3}$ $s^{-1}$ calculated, which is comparable to values reported in mountain and forest sites. Under*
*humid conditions the HONO source correlates well with NO and the authors attribute this missing HONO source to*
*emissions from soil. Finally, the impact of the HONO on OH production rates is calculated and the results show that*
*the HONO photolysis contributes, on average, 30% to OH production during the morning and evening.*
*Understanding the daytime source of HONO is important due to its role in OH formation and this study provides*
*important data on HONO sources in a location which is not strongly impacted by combustion sources.*
*The manuscript is well written, with appropriate sections and easy to follow. I recommend the manuscript for*
*publication in ACP after addressing the comments below:*

**Response:**
We thank the reviewer for the positive evaluation and please find our point-to-point responses as listed below.

**Specific comment:**
*One of the main concerns is that no uncertainty analysis has been performed for the HONO/NOx ratios or the*
*HONO budget and calculation of the missing HONO sources. This should include instrument uncertainties in the*
*HONO and NOx measurements along with errors in the PSS calculation. It would then be beneficial to include error*
*bars on Figure 5a and b, to show the upper and lower limits to the estimated unknown HONO source.*

**Response:**
Following the reviewer's suggestion, we now state all instruments' uncertainties in the revised manuscript. The
uncertainty of LOPAP (HONO) is 10%, based on the uncertainties of gas and liquid flow rates, regression of the
calibration curve, and calibration standard solutions (manual of LOPAP, QUMA 2004). The accuracy (2 sigma)
of the OH measurements was 29% and the precision (1 sigma) was $4.8x10^5$ molecules $cm^{-3}$ (personal contact with
Harder et al., hartwig.harder@mpic.de). The instrument uncertainties for NO, $NO_2$, $O_3$, J were already stated in
the original manuscript (20%, 30%, 5%, 10%, personal contact with Fischer et al, horst.fischer@mpic.de and
Crowley et al., john.crowley@mpic.de).
According to Gaussian error propagation, these instrument uncertainties affect the calculation of the unknown
HONO source $S_{HONO}$ with about 16%.
We agree that error bars would help to indicate the uncertainty of the source and sink terms in our calculations.
As Fig. 5 a and b show half-hourly mean values of diurnal patterns, we prefer to show the standard deviation of
the diurnal mean values as error bars, now included in Fig. 5 a and b, and to discuss the uncertainties in the text.

We now added in the revised versions of the manuscript:

page 4 line 13-15: "The accuracy of the HONO measurements was 10%, based on the uncertainties of liquid and
gas flow, concentration of calibration standard and regression of calibration ."

page 4 line 36-37: "The accuracy (2 sigma) of the OH measurements was 29% and the precision (1 sigma) was
$4.8x10^5$ molecules $cm^{-3}$."
page 10 line 21-23: "The uncertainty of the calculated missing source $S_{HONO}$ was estimated to be about 16%,
based on the Gaussian error propagation of instrument uncertainties of HONO, NO, $NO_2$, J, and OH."

in Fig. 5a/b error bars based on standard deviation of diel mean values are added

[Figure]

Revised Fig. 5a+b including error bars; as suggested in another comment, the NO₂ conversion rate for the
heterogeneous reaction in the dry case (b) is now adopted as suggested by the referee. (ΔHONO/Δt was added as
discussed in a comment by reviewer 2)

**Specific comment:**
*In section 5.1, the heterogeneous reaction of NO₂ to form HONO is estimated by applying an NO2-HONO*
*conversion rate of 1.6% h⁻¹ overnight. Under humid conditions the estimated values agree well with measured*
*values. A much lower rate of 0.22 % h⁻¹ was applied in the drier period, which the author's state matches better to*
*their observations. However, Fig 4. shows the measured HONO is still lower than the estimated values during some*
*periods overnight. Perhaps it would be better to determine a conversion rate under dry conditions for this site using*
*the NOx scaling approach (e.g. Sorgel et al., 2011) to compare with other studies, as I expect it is lower.*

**Response:**
Thank you for this comment and suggestion. Accordingly, we now use the approach from Alicke et al.,
2002+2003; Su et al. 2008b and Sörgel et al., 2011b for the conversion rate of the dry nights: $rate =$
$\frac{HONO_{t2}-HONO_{t1}}{\Delta t * NO_2}$ . The respective average conversion rate is now 0.36% h⁻¹(slightly higher than the 0.22 % h⁻¹ used
before).
Also, a more representative HONO nighttime starting concentration 4.4 ppt is now used to better match the
observations (as the original starting concentration 12.2 ppt was too high, the observed average concentration
decreased afterwards, see marked area in old fig 4b).

[Figure]

Upper panel of old Fig. 4b,

[Figure]

[Figure]

Upper panel of revised Fig. 4b with adopted nighttime $NO_2$ conversion rate based on Soergel et al. (2011b) and a more representative HONO nighttime starting concentration. According to comment below, error bars are now in darker color.

The manuscript, section "5.1 nighttime HONO accumulation" on page 8 from line 15-33 is now revised to:
"Instead, nighttime HONO concentrations can be estimated due to heterogeneous reaction of $NO_2$ described in Eq. (1) *(Alicke et al., 2002+ 2003; Su et al., 2008b; Sörgel et al., 2011b).* Three studies in different environments from a rural forest region in East Germany (Sörgel et al., 2011b) and a non-urban site in the Pearl River delta, China (Su et al., 2008b) to an urban, polluted site in Beijing (Spataro et al., 2013) found a conversion rate of *about* 1.6% $h^{-1}$ *(1.1-1.8 % $h^{-1}$).*

$$[HONO]_{het} = [HONO]_{evening} + 0.016 \text{ } h^{-1}[NO_2] \Delta t, \qquad\qquad\qquad (Eq. 1)$$

$[HONO]_{het}$ denotes the accumulation of HONO by heterogeneous conversion of $NO_2$, $[HONO]_{evening}$ the measured HONO mixing ratio at *20:30* LT, $[NO_2]$ the measured average $NO_2$ mixing ratio between *20:30* and 7:30 LT, $\Delta t$ time span in hours.

Measured and calculated HONO mixing ratios are compared in figure 4 (upper panel). During the humid period, during night the estimated (according Eq. (1), fig. 4a upper panel, grey line) and observed HONO mixing ratios are in good agreement ($R^2 = 0.9$). During the drier period the observed HONO mixing ratios were lower than the ones calculated with a $NO_2$ conversion rate of 1.6% $h^{-1}$. *Here the approach for the nighttime conversion frequency by e.g. Alicke et al., 2002+2003, Su et al., 2008b or Sörgel et al., 2011b ($rate = \frac{HONO_{t2}-HONO_{t1}}{\Delta t * NO_2}$) was used. The 7 days average conversion rate for the dry nights was 0.36% $h^{-1}$ (fig. 4b, upper panel, black line), comparable to results of Kleffmann et al. (2003) reporting a conversion rate of $6x10^{-7}$ $s^{-1}$ (0.22% $h^{-1}$) for rural forested land in Germany.*"

**comments:**
*Pg 3, L25-26. Please state the uncertainty of the HONO measurements here too.*

See comment above, the HONO uncertainty is now stated in this section.

*Pg 6, L18. The ± values in the parenthesis should be clarified. Are these 1-sigma standard deviation of the mean?*

Correct, this is now declared in the text on page 6 line 25: "…(± 25 pptv, 1σ standard deviation, following alike)"

*Pg 7, L7. It is stated that the mean NO mixing ratios are close to the detection limit at 2 pptv, however, this is actually below the detection limit, which is given as 5 pptv on Pg 4, L13.*

Sorry for this typo. Now the correct detection limit of 5 pptv of NO as written in the instrument description is now used here.

*Pg 8, L5-7. Here, HONO mixing ratios are estimated and compared to the measured HONO overnight using a conversion factor between NO2 and HONO of 1.6% h⁻¹. The authors cite three studies where this value has been determined, although, it should be made clear here that a range of values were reported across these studies.*

Correct. Now the range of values is stated in the modified version (see response on comment above) Sörgel et al. (2011) reported 1.1 ($\pm$0.65) % h⁻¹, Spataro et al. (2013) 1.5-1.8 % h⁻¹ and Su et al. (2008) came up with a best estimate of  1.6 % h⁻¹, based on different scaling methods.

*Pg 9, L25. Please state the values for $k_1$ and $k_2$ used in Eq. 2.*

The rate constants $k_1$, $k_2$ (and $k_3$ and $k_4$) are temperature dependent, so stating only one value would not be appropriate. The respective formulas were taken from Atkinson et al. (2004), as was already cited in the original manuscript.

$\text{NO+OH} \rightarrow k_1 = 7.4 \times 10^{-31}(\frac{T}{300})^{-2.4}[N_2]$

$\text{HONO+OH} \rightarrow k_2 = 2.5 \times 10^{-12}\exp(\frac{260}{T})$

$\text{NO+HO}_2 \rightarrow k_3 = 3.6 \times 10^{-12}\exp(\frac{270}{T})$

$\text{NO+O}_3 \rightarrow k_4 = 1.4 \times 10^{-12}\exp(-\frac{1310}{T})$

E.g. at a temperature of 23°C, typical for the measurement time on Cyprus: $k_1 = 1.36 \times 10^{-11}$ s⁻¹, $k_2 = 6.01 \times 10^{-12}$ s⁻¹, $k_3 = 8.96 \times 10^{-12}$ s⁻¹, $k_4 = 1.68 \times 10^{-14}$ s⁻¹

In the revised manuscript the temperature dependence is now pointed out, and respective numbers are given for a typical daytime temperature on Cyprus during the campaign (23°C)

*Fig 4: The error bars in figure 4b for the 0.2% rate are difficult to see, please use a darker color or use thicker lines.*

Thanks for indicating. Fig 4b is now changed accordingly (see response on comment above).

*In Figure 5, the caption states that a conversion rate of 1.6% h⁻¹ is used for $S_{Het\_NO2}$ , however, Figure 4b shows that a lower rate (0.22% h⁻¹) is more appropriate for the dry period. Please clarify which rate you use for Fig 5b.*

Correct. Indeed, in the original manuscript 1.6% h⁻¹ was used for both by mistake. In the revised manuscript the conversion rate adopted by Soergel et al. (2011b) is now used (0.36% h⁻¹; see comment above), and the figure caption corrected accordingly. We thank the reviewer for exposing this critical detail.

*Fig 6. Include units for $NO_2$ in the legend.*

Thanks, has been corrected accordingly.

*Fig. 4 and Fig 7. Please state in the figure captions what the error bars represent.*

The error bars represent one standard deviation of diel mean values. This is now specified in the figure captions of the revised manuscript (Fig. 4 and 7).

References
Sörgel, M., Trebs, I., Serafimovich, A., Moravek, A., Held, A., and Zetzsch, C.: Simultaneous HONO measurements in and above a forest canopy: influence of turbulent exchange on mixing ratio differences, Atmos. Chem. Phys., 11, 841-855, 2011(b)

Alicke, B., Platt, U., Stutz, J.: Impact of nitrous acid photolysis on the total hydroxyl radical budget during the limitation of oxidant production/pianura padana produzione di ozono study in Milan. Journal of Geophysical Research 107 (D22), 8196, 2002.

Alicke, B., Geyer, A., Hofzumahaus, A., Holland, F., Konrad, S., Paetz, H.W., Schaefer, J., Stutz, J., Volz-Thomas,
A., Platt, U.: OH formation by HONO photolysis during the BERLIOZ experiment. Journal of Geophysical Research
108 (D4), 8247, 2003.
Atkinson, R., Baulch, D. L., Cox, R. A., Crowley, J. N., Hampson, R. F., Hynes, R. G., Jenkin, M. E., Rossi, M. J.,
and Troe, J.: Evaluated kinetic and photochemical data for atmospheric chemistry: Volume I - gas phase reactions of
O-x, HOx, NOx and SOx species, Atmospheric Chemistry and Physics, 4, 1461-1738, 2004.
Su, H., Cheng, Y. F., Cheng, P., Zhang, Y. H., Dong, S., Zeng, L. M., Wang, X., Slanina, J., Shao, M., and
Wiedensohler, A.: Observation of nighttime nitrous acid (HONO) formation at a non-urban site during PRIDE-
PRD2004 in China, Atmospheric Environment, 42, 6219-6232, 2008(b).
Spataro, F., Ianniello, A., Esposito, G., Allegrini, I., Zhu, T., and Hu, M.: Occurrence of atmospheric nitrous acid in
the urban area of Beijing (China), The Science of the total environment, 447, 210-224, 2013.

**Anonymous Referee #2**

**General comments:**
*This paper uses measurements of HONO with a wide range of supporting data to assess sources of HONO in the*
*remote coastal site in Cyprus. The findings are that there is a common source of HONO and NO and it is speculated*
*that this is emission from microbial communities on soil surfaces. The work is important as HONO provides a route*
*to OH radicals that is often not considered and sources of HONO in both urban and remote regions are uncertain.*
*The authors have done a good job presenting their data and the conclusions they draw are reasonable. It*
*undoubtedly adds to the sphere of knowledge surrounding atmospheric HONO. The paper is well presented with*
*good clear figures and should be published in ACP subject to the authors addressing the following comments.*

**Response:**
We thank the reviewer for the positive feedback. Please find our point-to-point responses listed below.

**Specific comment:**
*The main conclusion of the paper is that there is a soil source of HONO and NO, which is arrived upon by looking at*
*correlations between the 'missing' HONO source (i.e. the difference between HONO calculated using a steady state*
*approximation including a series of known sources and the measured HONO) and a missing source of NO (based on*
*NO deviations from the Leighton ratio). A strong correlation is given as evidence of a common source. Is this source*
*thought to be photolytically driven? If not why are observations of NO at night seemingly zero (although it is quite*
*difficult to see the exact levels on the plots), whereas HONO is shown to increase during the night. Maybe this is just*
*a result of NO reacting with $O_3$ before the measurement location but the authors should clarify this.*

**Response:**
True, the correlation analysis was based on daytime values only. For nighttime conditions, the chemistry is
relatively slow and transport processes could strongly influence the budget of nitrogen-containing species, that's
why we focus on the daytime chemistry.
The difference in nighttime accumulation of NO and HONO may be due to other reasons, like $NO_2$
heterogeneous conversion, being relevant for HONO accumulation within a shallow nocturnal boundary layer
(here 0.4-1.6 %, in line with other literature, see chapter "5.1 nighttime HONO accumulation"), while there is no
chemical source for NO. Also the nighttime reaction of NO + OH forming HONO would result in a preference in
HONO accumulation, with nighttime OH concentration sometimes as high as $1 \times 10^6$ molecules $cm^{-3}$ (see Fig. S3).

As suggested by the referee, NO titration by the reaction with $O_3$ may also play a role for the absence of nighttime NO accumulation, with continuously high $O_3$ concentrations (60-90 ppb), while there is no major loss of HONO due to the lack of photolysis. Another option would be different temperature dependencies of NO and

HONO emissions from soil (e.g., Oikawa et al., 2015; Mamtimin et al., 2016, which is now stated on page 12 line

31-35).

We modified one sentence of the "result" chapter, page 7, line 22). "In the absence of local NO sources low nighttime values are a result of the conversion of NO to $NO_2$ by $O_3$ which was continuously high (Beygi et al.,

2011)"

**Specific comment:**

*How far from the potential soil emission source in the measurement site? The authors should also comment on how*

*this effects the validity of the steady state approximation, with reference to the Lee et al. 2013 study that gives*

*caveats for the use of a steady state approximation to interpret HONO measurements.*

**Response:**

Thanks to the comment. We are afraid that we may not have described our calculation properly and the using of

$HONO_{PPS}$ was misleading. We actually followed the method in Su et al. (2008a) to calculate HONO missing source. With this method, we did not assume HONO to be at PSS, because the measured d[HONO]/dt has been accounted in the HONO missing source estimation and according to our measurement d[HONO]/dt was not equal to zero, which did mean that HONO was not at PPS.

$$S_{HONO} = J_{HONO}[HONO] + k_2[OH][HONO] - k_1[OH][NO] - k_{het}[NO_2] + \frac{\Delta[HONO]}{\Delta t}$$

Lee et al. (2013) states that assuming HONO to be completely at PSS will likely overestimate the strength of any

"unknown source". In the Lee et al. (2013) case, the authors would come up with up to 1.1 ppb $h^{-1}$ with the PSS

approach. They argue that instead of presuming a PSS, they can explain the observed HONO from pure precursor chemistry by applying a simple chemical box model.

Lee et al. (2013) argue that the PSS assumption might not have been valid for their case study, because the transport time from nearby $NO_x$ vehicle exhaust emission sources to the measurement site was likely less than the time required for HONO to reach PSS. Using a chemical box model, Lee et al. (2013) demonstrated that there is initially net HONO formation from assumed strong emissions (100 ppm for the sum of NO, $NO_2$, HONO), as high levels of NO in vehicle exhaust react with assumed entrainment of ambient OH. In the respective model daytime show-case, this HONO net production (d[HONO]/dt>0) , sustained for 2.5 min after precursor emission.

Subsequently, net HONO loss dominated by photolysis led to a negative d[HONO]/dt in their calculations, which was sustained for several minutes until PSS is established after ca. 10 min (for mean daytime conditions) or up to several hours depending on time of day. This way, Lee et al. (2013) claim that "for all conditions, d[HONO]/dt is negative for a specific period of time, during which sampling vehicle exhaust can lead to overestimates of secondary HONO sources *if a photostationary state is inappropriately assumed*", and hence "… there exists a window of time in which d[HONO]/dt is negative. Erroneously assuming the presence of PSS during this time period would lead to overestimates of secondary HONO sources."

With respect to our analysis, first of all, we did not assume that HONO PSS is fully established at our measurement site (d[HONO]/dt was not equal to 0). However, even though for the Cyprus case, the mean upwind distance between the measurement site and the coast line is about 6 km. With a mean wind velocity of 3 m $s^{-1}$ the respective air mass travel time over land/soil surface is about half an hour, i.e., several times the daytime lifetime of HONO. Moreover, and in strong contrast to Lee et al. (2013), at the Cyprus site the concentrations of HONO

precursors were extremely low. In the Cyprus case, the observed atmospheric load of precursors (NO and OH) is by far too low to explain the observed HONO concentrations, or d[HONO]/dt, respectively (see Fig. 5). Even doubling the contribution of the chemical source (NO + OH) would not lead to a substantial reduction of the strength of the calculated un-identified HONO source.

To account for any caveats of any PSS assumptions, we now state in the text (page 10, line 14-30):

"Lee et al. (2013) argue that the HONO PSS assumption might overestimate the strength of any un-identified source, if the transport time from nearby NOx emission sources to the measurement site is less than the time required for HONO to reach PSS. In this study, the missing source was calculated according to Su et al., 2008a (eq.3), where PSS was not assumed. Also in our measurements, dHONO/dt was not equal to zero, as HONO was not at PSS.

$$S_{HONO} = J_{HONO}[HONO] + k_2[OH][HONO] - k_1[OH][NO] - k_{het}[NO_2] + \frac{\Delta[HONO]}{\Delta t} \qquad \text{(Eq.3)}$$

with [HONO] being the measured HONO concentration and $k_{het}$ the heterogeneous conversion rate of $NO_2$ to HONO, which was discussed above to be 1.6% $h^{-1}$ during the wet period and 0.36% $h^{-1}$ during the dry period. $\Delta[HONO]/\Delta t$ is the observed change of HONO concentration unequal to 0. The uncertainty of the calculated missing source $S_{HONO}$ was estimated to be about 16% based on the Gaussian error propagation of instrument uncertainties of HONO, NO, $NO_2$, J and OH.

Nevertheless, at the study site of Cyprus, the mean upwind distance between the measurement site and the coast line was about 6 km, and the mean wind velocity was about 3 m $s^{-1}$. Accordingly, the respective air mass travel time over land is estimated to be about half an hour, which is somewhat longer than the daytime lifetime of HONO and might provide enough time for the equilibrium processes. Furthermore and in a strong contrast to Lee et al. (2013), at the Cyprus site the concentrations of HONO precursors (NO and OH) were extremely low, by far too low to explain the observed HONO concentrations."

**Specific Comment:**
*I find the analysis of OH production showing the importance of HONO confusing because it details production of OH from HCHO, which is indirect and requires conversion of the $HO_2$ produced with NO to form OH. I believe it would be better to just include $HO_2$ + NO as an OH source, regardless of where the $HO_2$ is coming from. Another option would be to have a total HOx radical budget analysis.*

**Response:**
We thank the referee for disclosing this critical detail. OH budget analysis including photolysis of HCHO was done before e.g. by Alicke et al., 2002 but then also $RO_2$ primary production should be considered as it will also be converted in OH through cycling processes. HONO photolysis, ozonolysis of alkenes and photolysis of $O_3$ and subsequent reaction with water contribute to the primary OH production. HCHO photolysis firstly forms $HO_2$ which is in fast equilibrium with OH e.g. via the reaction of $HO_2$ and NO. Therefore it contributes to secondary OH production. In this study we focus on the evaluation of HONO sources and wanted to give a brief outlook on its importance on OH. To realize this we just show the primary OH production routes (and deleted the OH production via HCHO photolysis). Furthermore we changed the term "OH production" into "primary OH production".
As written below, a complete detailed HOx budget analysis will be published from colleagues soon.

[Figure]

Revised fig. 9: Average diel pattern of primary OH production from HONO, $O_3$ and VOC, shown as a) production rate and b) percentage contributions to primary OH production.

**Specific Comment:**

*The authors should also comment on the fact that the HONO source here is only important near to the surface (an estimate could be made of the vertical structure of HONO) and thus it is not relevant for the entire troposphere. This is important when considering HONO as an atmospheric 'oxidant'.*

**Response:**

Indeed, many studies have shown decreasing HONO mixing ratios with altitude in the lowest few hundred meters of the troposphere (Vogel et al., 2003; Zhang et al., 2009; Young et al., 2012; Wong et al., 2012 and 2013; VandenBoer et al., 2013). According to the modelling results of Wong et al. 2013, we estimate that the ground HONO source could be important for up to 200–300 m a.g.l. According to the referee's suggestion we now state in the introduction (Page 3, line 9-13):

"Many studies have shown decreasing HONO mixing ratios with altitude in the lowest few hundred meters of the troposphere, due to respective short atmospheric lifetime compared to vertical transport time (Wong et al., 2012 and 2013; Vogel et al., 2003; VandenBoer et al., 2013; Zhang et al., 2009; Young et al., 2012; ). According to the modelling results of Wong et al. 2013, we estimate that the ground HONO source could be important for up to 200–300 m a.g.l. This indicates that HONO is more relevant for the OH budget close to the surface than in high altitude air masses."

**Comment:**

*The authors mention in the experimental description that OH was measured during the field campaign but there is then no further mention of it in the manuscript. Have the authors (or anyone else) examined the OH data to assess if the measured HONO is required to close the HOx budget? I realise this may be the subject of further publications but if it is stated that HOx was measured it seems odd that no mention is made of the results.*

**Response:**

We thank the referee for this suggestion. In this study, we focused on HONO and its missing daytime source. OH data were used to calculate the HONO budget and $HO_2$ data were used to study the NO budget. The potential contribution of HONO photolysis to OH production is studied in a short chapter showing different OH production routes. A "total $HO_x$ radical budget analysis" is not focus of this manuscript. A future CYPHEX paper of colleagues will deal with the $HO_x$ budget closure study, including a detailed box modelling approach for the total $HO_x$ budget and OH recycling. The manuscript will be submitted soon.

**Comment:**

*The manuscript is generally well referenced however a recent study by Lee et al. 2016 in London contains a lot of detail about potential HONO sources in an urban area and should be referenced. There is also a recent study by Mamtimin et al. (2016) which discusses biogenic NO and HONO emissions that seems to be extremely relevant to this work. The authors should comment on how their results compare to this.*

**Response:**

We greatly appreciate these reference suggestions. Both fit well into this study.

Mamtimin et al., 2016 is now cited in the introduction (page 3, line3), and twice in "common daytime source of HONO and NO" (page 12, line 28 and line 31-35) when comparing with other studies on HONO and NO emission. "Mamtimin et al. (2016) investigated HONO and NO emissions of natural desert soil and with grapes or cotton cultivated soils in an oasis in the Taklamakan desert in the Xinjiang region in China. After irrigation they didn´t find direct emission, but when the soil had almost dried out (gravimetric soil water content 0.01-0.3) emissions up to 115 ng N m$^{-2}$ s$^{-1}$ were detected. In addition they observed soil-temperature dependent emission of reactive nitrogen.

Lee et al. (2016) is now cited once in the introduction (page 3, line 17) and in "daytime HONO budget" (page 11, line 15-16), discussing possible light induced HONO formation and comparing correlation factors:

"Lee et al. (2016) found even lower correlation with [$NO_2$] ($R^2 = 0.0001$) but similar good correlation with $J_{NO2}*$[$NO_2$] ($R^2=0.70$) at an urban background site in London."

**Minor comments:**

*The authors should make sure they clarify what the error bars on plots and in the text actually refer to (e.g. figures 4 and 7)*

Correct. As also suggested by Referee #1, we now clarify what the error bars on plots and in the text actually refer to. Error bars in Fig. 4 and 7 (and ± values) indicate standard deviation (1 sigma).

*P. 12 line 11: Use $O_3$ rather than ozone as has been done in the rest of the manuscript*

Thanks for noticing. Is now changed accordingly in the text.

*P. 7 line 7: there is a discrepancy between the detection limit stated here (2pptv) and that in the experimental section (5pptv) – please confirm.*

Thanks for indicating. As also suggested by referee #1, a detection limit of 5 pptv is now stated in the revised versions of the manuscript.

[revised manuscript text omitted]

[c]  poorly correlated R² > 0.5

*  anti-correlated

| | during the humid period | | | | during the dry period | | | |
|---|---|---|---|---|---|---|---|---|
| | | | Time of day average | | | | Time of day average | |
| | HONO | $S_{HONO}$ | HONO | $S_{HONO}$ | HONO | $S_{HONO}$ | HONO | $S_{HONO}$ |
| T | 0.006 | 0.1216 | 0.031 | 0.1123 | 0.120 | 0.0163 | 0.453 | -0.00415 |
| RH | 0.000 | 0.08192* | 0.010* | 0.12746* | 0.374 | 0.193227 | **0.730**[b] | 0.6083[cb] |
| Heat flux | 0.110 | 0.2743 | 0.184 | 0.55491[c] | 0.502[c] | 0.3035 | **0.685**[b] | 0.59634[c] |
| $J_{NO2}$ | 0.150 | 0.4657 | 0.245 | **0.69869**[b] | **0.678**[b] | 0.32057 | **0.829**[a] | **0.65764**[b] |
| NO | 0.168 | 0.18835 | 0.418 | **0.65076**[b] | 0.487 | 0.32301 | **0.730**[b] | 0.34029 |
| $NO_2$ | 0.066 | 0.0675 | 0.300 | 0.353267 | 0.037 | -0.0023* | 0.619[c] | 0.1741 |
| $NO_2$*RH | 0.084 | 0.04853 | 0.294 | 0.171245 | 0.161 | 0.0210 | **0.714**[b] | 0.45236[e] |
| $NO_2$*RH*aerosol surface | 0.047 | 0.0729 | 0.111 | 0.250147 | 0.241 | 0.10685 | 0.557[c] | 0.62551[c] |
| $NO_2$*J | 0.214 | 0.2691 | 0.427 | **0.910845**[a] | 0.358 | 0.0168 | **0.872**[a] | **0.65703**[b] |
| $NO_2$*RH*J | 0.231 | 0.24471 | 0.467 | **0.850775**[b] | 0.434 | 0.0685 | **0.820**[a] | **0.7703**[b] |
| $NO_2$*RH*J*aerosol surface | 0.140 | 0.15260 | 0.465 | **0.78495**[b] | 0.414 | 0.17130 | **0.664**[b] | 0.67831[cb] |
| $S_{NO}$ | | 0.294323 | | **0.77820**[b] | | 0.00359* | | -0.0094* |

[Figure]

**Figure 1: Map of location: the red star shows the location of Ineia and the measuring site. The four red points mark the main cities of Cyprus, Nicosia, Larnaca, Limassol and Paphos (clockwise ordering), map produced by the Cartographic Research Lab University of Alabama, map of Cyprus: google maps.**

[Figure]

**Figure 2: Airflow conditions during the CYPHEX campaign: a) Measured local wind direction, b) back trajectories calculated with NOAA Hysplit model showing examples for the two main air mass origins (48 hours, UTC = LT - 3 h).**

[Figure]

**Figure 3: Measured variables during the whole campaign from 7th July to 4th August 2014, a) meteorological data (Temperature T, relative humidity RH, wind direction and speed wd, ws) and O₃ and CO indicating stable conditions, in the lower panel the bar indicates the air mass origin: bright blue = westerly, brownish = northerly, b) observed mixing ratios of HONO, NO₂ and NO, and the photolysis frequency J$_{HONO}$ and the HONO/NOx ratio. The yellow and blue boxes reflect the dry and the humid periods, respectively.**

[Figure]

[Figure]

**Figure 4: Diel variation of meteorological data (Temperature T, relative humidity RH), NO and NO$_2$ mixing ratios, the photolysis rate for HONO J$_{HONO}$ and HONO mixing ratios (pink: measured, violet: daytime photostationary state PSS, grey: nighttime heterogeneous NO$_2$ conversion) and HONO/NOx ratio for a) average for period when RH was above 60% (blue box in Fig. 3) and b) average for dry period when RH was below 60% (yellow box in Fig. 3). Error bars represent standard deviation of diel mean.**

**Kommentar [HM1]:** Corrected according to reviewer 1 (comment 2): changed "starting" concentration for het. Reaction calculation (both rates); (comment 8) error bars of 0.2% rate are black, and now very small-> so not possible to make it more clear

[Figure]

**Figure 5: HONO budget analysis for a) the humid and b) the dry period. $S_{OH+NO}$ (black) stands for the formation rate of HONO via the reaction of NO and OH, $S_{Het\_NO2}$ (yellow) is the formation rate for the heterogeneous reaction of $NO_2$ (conversion rate a=1.6% h$^{-1}$; b=0.36% h$^{-1}$), $L_{phot}$ (green) and $L_{OH+HONO}$ (blue) are the loss rates via photolysis and the reaction with OH and $S_{unknown}$ is the unknown source. Error bars indicate standard deviation of diel mean.**

[Figure]

**Figure 6: NO₂ (color-coded) and RH dependence of the sources of NO (S$_{NO}$) and HONO (S$_{HONO}$).**

[Figure]

**Figure. 7: Diel profile of both unknown sources $S_{HONO}$ (a) and $S_{NO}$ (b) for all data, humid (excluding transition days: 25.7. and 2.8 and 15.7. as RH conditions changed too quickly) and dry periods. Error bars indicate standard deviation of diel average.**

[Figure]

[Figure]

**Figure 8: Correlation of $S_{HONO}$ to light induced $NO_2$ reaction (for both periods, humid = blue triangle, dry = orange square), to NO and $S_{NO}$ (only for humid period, excluding the 3 days mentioned above); time of day average data were used.**

[Figure]

**Figure. 9: Average diel pattern of primary OH production from HONO, O₃, HCHO and VOCalkenes, a) shown as a) production rate and b) percentage contributions to totalprimary OH production.**